



# Treatment of non-ideality in the multiphase model SPACCIM-Part2: Impacts on the multiphase chemical processing in deliquesced aerosol particles

5   **Ahmad J. Rusumdar[1,2], Andreas Tilgner[1], Ralf Wolke[1], and Hartmut Herrmann[1]**

[1] Leibniz Institute for Tropospheric Research (TROPOS), Leipzig, 04318, Germany

[2] Now at: FERCHAU Engineering, Niederlassung Karlsruhe, Karlsruhe 76185, Germany

*Correspondence to:* Ralf Wolke (wolke@tropos.de) and Hartmut Herrmann (herrmann@tropos.de)





## Abstract

Tropospheric deliquesced particles are characterised by concentrated non-ideal solutions ('aerosol liquid water' or ALW) that can affect the occurring multiphase chemistry. However, such non-ideal solution effects have generally not yet been considered in and investigated by current complex multiphase chemistry models in an adequate way. Therefore, the present study aims at accessing the impact of non-ideality on multiphase chemical processing in concentrated aqueous aerosols. Simulations with the multiphase chemistry model (SPACCIM-SpactMod) are performed in different environmental and microphysical conditions with and without a treatment of non-ideal solutions in order to assess its impact on aqueous-phase chemical processing.

The present study shows that activity coefficients of inorganic ions are often below unity under 90 % RH-deliquesced aerosol conditions, and that most uncharged organic compounds exhibit activity coefficient values of around or even above unity. Due to this behaviour, model studies have revealed that the inclusion of non-ideality considerably affects the multiphase chemical processing of transition metal ions (TMIs), oxidants, and related chemical subsystems such as organic chemistry. In detail, both the chemical formation and oxidation fluxes of $Fe(II)$ are substantially lowered by a factor of 2.8 in the non-ideal base case compared to the ideal case. The reduced $Fe(II)$ processing in the non-ideal base case, including lowered chemical fluxes of the Fenton reaction (-70 %), leads to a reduced processing of $HO_x/HO_y$ under deliquesced aerosol conditions. Consequently, higher multiphase $H_2O_2$ concentrations (larger by a factor of 3.1) and lower aqueous-phase OH concentrations (lower by a factor of $\approx 4$) are modelled during non-cloud periods. For $H_2O_2$, a comparison of the chemical reaction fluxes reveals that the most important sink, the reaction with $HSO_3^-$, contributes with a 40 % higher flux in the non-ideal base case than in the ideal case, leading to more efficient sulfate formation. On the other hand, the chemical fluxes of the OH radical are about 50 % lower in the non-ideal base case than in the ideal case, including lower degradation fluxes of organic aerosol components. Thus, considering non-ideality influences the chemical processing and the concentrations of organic compounds under deliquesced particle conditions in a compound-specific manner. For example, the reduced oxidation budget under deliquesced particle conditions leads to both increased and decreased concentration levels, e.g. of important $C_2/C_3$ carboxylic acids. For oxalic acid, the present study demonstrates that the non-ideality treatment enables more realistic predictions of high oxalate concentrations than observed under ambient highly polluted conditions. Furthermore, the simulations implicate that lower humidity conditions, i.e. more concentrated solutions, might promote higher oxalic acid concentration levels in aqueous aerosols due to differently affected formation and degradation processes.



# 1    Introduction

Aerosol particles represent a more or less omnipresent multiphase compartment within the troposphere, which generally comprises complex mixtures containing different organic and inorganic compounds including water (see e.g. Saxena and Hildemann (1996); Pöschl (2005); Hallquist et al. (2009); and references therein). Aerosol properties and their impacts largely depend on chemical composition, size, and phase. Due to the various important impacts of aerosols on atmospheric chemistry (Andreae and Crutzen, 1997; Ravishankara, 1997), air pollution (Akimoto, 2003; von Schneidemesser et al., 2015), biosphere (Adriano and Johnson, 1989), climate (Charlson et al., 1992; Lohmann and Feichter, 2005; Boucher et al., 2013; Myhre et al., 2013), and public health (Holgate, 1999; Brunekreef and Holgate, 2002; Lelieveld et al., 2015), it is a key challenge to understand how aerosol particles are physically and particularly chemically processed in the tropospheric multiphase system in order to finally clarify their global importance and impacts.

Tropospheric aerosol particles are a complex multiphase and multi-component environment, in which a variety of physical and chemical processes can alter the physical and chemical composition of the troposphere, potentially on a global scale. With regard to aerosol particles, chemical reactions can occur heterogeneously at the surface and homogenously in a bulk of organic and aqueous aerosol particles (Ravishankara, 1997; George et al., 2015; Herrmann et al., 2015). Through interaction with ambient water vapour, aerosol particles can be deliquesced, forming an aqueous-aerosol phase characterised by highly concentrated solutions with rather low aerosol liquid water contents (ALWCs) of between $3 \cdot 10^{-7}$ and $1 \cdot 10^{-3}$ g m$^{-3}$ (Herrmann et al., 2015). Chemical bulk processes in the deliquesced aerosol phase are expected to be of importance (Ervens et al., 2011; Tilgner et al., 2013; Herrmann et al., 2015). Aqueous-phase processes are known to be very efficient in occurring on short timescales and in producing secondary compounds, i.e. contributing to secondary aerosol mass, e.g. aqSOA (Tilgner and Herrmann, 2010; Ervens et al., 2011; McNeill et al., 2012a; Tilgner et al., 2013; Ervens, 2015; Herrmann et al., 2015). Additionally, chemical processes in aqueous aerosols can alter the composition of the surrounding gas phase (Tilgner et al., 2013). However, compared to processes in well-diluted environments such as the tropospheric gas and cloud phase, chemical processes in highly concentrated non-ideal solutions of deliquesced aerosol particles are much less investigated.

To access the role of multiphase chemical processes related to aerosol particles and cloud droplets, a variety of complex multiphase chemistry mechanisms and multiphase models were developed and applied in the past (see e.g. (Sander and Crutzen, 1996); Vogt et al. (1996); Vogt et al. (1999); von Glasow et al. (2002a, 2002b); Deguillaume et al. (2004); Ervens et al. (2004); Lim et al. (2005); Barth (2006); Pechtl et al. (2007); Tilgner and Herrmann (2010); Ervens et al. (2011); Ervens (2015); Hoffmann et al. (2016); Mouchel-Vallon et al. (2017); Hoffmann et al. (2018); Rose et al. (2018)). Many studies focused on the role of chemical aqueous-phase processes mainly involving cloud droplets and partly deliquesced particles. Model studies dealing with chemical processes in deliquesced particles were often focused on marine particles (see e.g., Vogt et al. (1996); Vogt et al. (1999); von Glasow and Sander (2001); von Glasow et al. (2002a, 2002b); von Glasow et al. (2004); Pechtl et al. (2007); Bräuer et al. (2013); Hoffmann et al. (2016)) and to a minor fraction on continental particles (Tilgner and Herrmann, 2010; McNeill et al., 2012b; Tilgner et al., 2013; Guo et al., 2014). Such studies often revealed the potential role





of deliquesced particles to act as a reactive aqueous chemical environment in the troposphere (see e.g., Tilgner et al. (2013)). However, the treatment of particle-phase chemistry in complex chemistry box models is mostly approximated by dilute electrolyte solution, neglecting non-ideal solution effects. However, because of very low ALWCs, deliquesced aerosol solutions are often characterised by quite high ionic strengths of 2 - 45 mol L$^{-1}$ (see Herrmann et al. (2015) and references

therein). Therefore, highly concentrated solutions of typical deliquesced particles cannot be treated as ideal solutions anymore, where intermolecular interaction forces between the non-solvent molecules are almost unimportant. Laboratory experiments (see e.g., Cappa et al. (2008)) have also confirmed that mixtures of aerosol components which are solids as pure substances become liquid-like and exhibit non-ideal behaviour.

In highly concentrated solutions, dissolved electrolytes and polar non-electrolytes are located very close to each other, causing

electrostatic forces or other physical interactions. These intermolecular forces are able to influence both the phase transfer processes of a compound and its affinity to undergo chemical reactions. Hence, an adequate treatment of non-ideality is definitely needed for simulating chemical processes in deliquesced particles. Thermodynamic non-ideality in a specific multicomponent mixture, caused by all-molecular interactions, can be represented by activity coefficients of the different components. Consequently, suitable chemistry models have to apply activities instead of concentrations in multiphase

chemistry models. For the sake of completeness, it is noted that besides the thermodynamic non-ideality, ionic strength effects of reactions also play a role in highly concentrated solutions (see Herrmann et al. (2015) for further details).

In the past, several approaches have been developed for the computation of required activity coefficients (see e.g. Pitzer (1991); Li et al. (1994); Prausnitz et al. (1998); Yan et al. (1999); Ming and Russell (2002); Raatikainen and Laaksonen (2005); Zaveri et al. (2005a); Zaveri et al. (2005b); Erdakos et al. (2006a); Erdakos et al. (2006b); Clegg et al. (2008); Zuend et al. (2008);

Zuend et al. (2011)). At the same time, several authors attempted to develop these activity coefficient models to simulate aerosol thermodynamic equilibrium at variable complexity (e.g., AIM (Clegg et al., 1998a, b), GFEMIN (Ansari and Pandis, 1999), ISORROPIA and ISORROPIA II (Nenes et al., 1998; Fountoukis and Nenes, 2007), EQUISOLV II (Jacobson et al., 1996; Jacobson, 1997); MESA (Zaveri et al., 2005a); UHAERO (Amundson et al., 2006; Amundson et al., 2007)). However, complex multiphase chemistry models dealing with deliquesced particles usually neglect or only roughly estimate the effect

of solution non-ideality on chemical processing (see e.g., Vogt et al. (1999); von Glasow et al. (2002a, 2002b); Tilgner and Herrmann (2010); Bräuer et al. (2013); Mao et al. (2013); Tilgner et al. (2013); Guo et al. (2014)). Therefore, detailed studies characterising the effect of non-ideal solutions on multiphase chemistry, e.g. in tropospheric deliquesced particles, are still lacking. For this reason, during the last years considerable effort has been devoted to developing kinetic model frameworks for the modelling of processes in multicomponent atmospheric particles, which include both a detailed description of organic

and inorganic multiphase chemistry as well as detailed thermodynamic comprehensions of its non-ideal behaviour (see Shrivastava et al. (2011); Rusumdar et al. (2016)).

In this context, the SPACCIM model framework (Wolke et al., 2005) was advanced by implementing a complex activity coefficient calculation module. The extended model approach of SPACCIM, including the treatment of non-ideality, is



described in the companion paper (see Rusumdar et al. (2016)). In the past, SPACCIM has been successfully applied in several chemical process model studies using the complex multiphase mechanism CAPRAM (Herrmann et al., 2005; Tilgner and Herrmann, 2010; Bräuer et al., 2013; Tilgner et al., 2013), focusing on both chemical and microphysical processes in cloud droplets and deliquesced particles, assuming ideal solution conditions. In the companion paper, the considered module

SpactMod (Rusumdar et al., 2016) was tested against other activity coefficient modules and compared to the measurement data as well as showed its applicability within the model framework. Consequently, the present follow-up study aims at investigating and, finally, assessing the impact of non-ideality on aqueous-phase chemical processing in tropospheric deliquesced particles. Overall, the treatment described in Rusumdar et al. (2016) and applied here makes the utilisation of CAPRAM in aerosol liquid water (ALW) chemistry possible within its full scope of details.

This paper is split into four sections. Sect. 2 outlines the multiphase model framework SPACCIM-SpactMod and the applied multiphase chemical mechanism along with performed simulations. In Sect. 3, the modelled activity coefficients of inorganic and organic compounds under different environmental conditions are discussed separately. Subsequently, the modelled results are described and discussed, including the differences between the simulations, considering multiphase chemistry as ideal and non-ideal solutions for key chemical organic and inorganic subsystems. Finally, the main findings of the study are summarised

in Sect. 4 and an outlook on future model developments and investigations is given.

## 2    Model and mechanism description

### 2.1   Multiphase chemistry model SPACCIM-SpactMod

In the present study, the extended version of the multiphase chemistry model SPACCIM (Wolke et al., 2005), including the considered activity coefficient module (SpactMod, Rusumdar et al. (2016)) entitled as SPACCIM-SpactMod in the following,

has been applied to investigate the influence of the treatment of non-ideality on multiphase chemistry.

To put it briefly, SPACCIM is an air parcel model combining a complex size-resolved multiphase chemistry model and a cloud microphysical model. The interaction between both models is implemented by a coupling scheme, enabling both models to run separately as far as possible. Changes of the chemical aerosol composition due to gas scavenging and chemical reactions have a feedback on the microphysical processes and properties. The multiphase chemical model uses a high-order implicit

time integration scheme, which exploits the special sparse structure of the model equations (Wolke and Knoth, 2002). The microphysical model applied in the SPACCIM model framework is based on the work of (Simmel and Wurzler, 2006) and Simmel et al. (2005). In the microphysical model, the growth and shrinking processes of aerosols by water vapour diffusion as well as nucleation and growth/evaporation of cloud droplets and other microphysical processes, such as impaction of aerosol particles and collision/coalescence of droplets, are described explicitly. Based on the Köhler theory (Köhler, 1936), the

dynamic growth rate in the condensation/evaporation process and the droplet activation is implemented. Moreover, cloud droplet formation, evolution, and evaporation are implemented using one-dimensional sectional microphysics, considering



deliquesced particles and droplets, respectively. Due to the focus of present model studies on the effects of non-ideality on multiphase chemistry, microphysical processes, such as impaction of aerosol particles, collision/coalescence of droplets and precipitation, are not covered in the model of the present study. Overall, the complex model framework enables detailed investigations of multiphase chemical processing of gases, deliquesced particles, and cloud droplets. Further details about the

SPACCIM model are given elsewhere in the literature (see Sehili et al. (2005); Wolke et al. (2005) and references therein).

In its latest version, SPACCIM-SpactMod additionally considers the treatment of non-ideality. In the incorporated activity coefficient module SpactMod (Rusumdar et al., 2016), non-ideality is treated with the approach by Zuend et al. (2008)/(Zuend et al., 2011) applied in the AIOMFAC model (Aerosol Inorganic–Organic Mixtures Functional groups Activity Coefficients, http://www.aiomfac.caltech.edu/index.html). This model is a thermodynamic group-contribution model specifically developed

to describe typical tropospheric aerosol compositions. The group-contribution concept treats organic molecules as structures composed of different functional groups. This approach allows for the representation of thousands of different organic compounds using a relatively small number of functional groups. The module SpactMod is based on AIOMFAC and combines a Pitzer-like model approach with a UNIFAC model to predict activity coefficients of mixed organic-inorganic systems. The activity coefficient model approach considers both ion interactions and interactions of organic compounds.

The non-ideality of a thermodynamic system is characterised by excess Gibbs energy, which is expressed as the sum of long-range (LR), middle-range (MR), and short-range (SR) contributions. The LR interactions are described by an extended Debye–Hückel term. The MR part represents the effects of interactions involving ions and permanent or induced dipoles, and it contains most of the adjustable parameters to describe concentrated aqueous electrolyte solutions and organic–inorganic mixtures. The SR term is calculated through a modified UNIFAC approach (Fredenslund et al., 1975) using the revised

parameter set of Hansen et al. (1991). Modifications of the UNIFAC model part within SpactMod further the introduction of inorganic ions in order to account for their effects on the entropy and enthalpy of mixing apart from their charge-related interactions. The original SpactMod (Rusumdar et al., 2016) mainly applies the interaction parameter data set of Zuend et al. (2008). In order to treat various aerosol constituents, additional parameters were included from the modified LIFAC approach by Kiepe et al. (2006), which can be rewritten in the AIOMFAC formalism. For the present study, the module has been updated

and extended based on Zuend et al. (2011).

Overall, SpactMod ensures a reliable calculation of activity coefficients for organic-electrolyte mixtures from diluted aqueous solutions to mixtures of high ionic strength composed of various ions and organic compounds with several functional groups. A detailed description of SpactMod and its integration into SPACCIM is given in a companion paper (Rusumdar et al., 2016). Lastly, it has to be noted that the current version of SpactMod does not consider the latest development stage of the AIOMFAC

model (Ganbavale et al., 2015; Gervasi et al., 2019). Nevertheless, it has also been observed that SPACCIM-SpactMod allows for model simulations with and without consideration of non-ideality by treating molarities as concentrations and activities, respectively. Therefore, the present implementation of the SPACCIM-SpactMod allows for detailed studies investigating the importance of non-ideality effects on the multiphase chemistry in deliquesced particles for the first time.





## 2.2 Multiphase chemistry mechanism

For current model applications, the existing aqueous-phase mechanism CAPRAM 3.0i (Herrmann et al., 2005; Tilgner and Herrmann, 2010; Tilgner et al., 2013) and the latest CAPRAM4.0 (Bräuer et al., 2019) were rather complex. Hence, a condensed chemical mechanism that still reflects the main chemical processes in tropospheric cloud droplets and deliquesced particles was extracted from the CAPRAM3.0i mechanism. The current employed mechanism consists of the inorganic core of the CAPRAM3.0i (Ervens et al., 2003; Herrmann et al., 2005; Tilgner et al., 2013) combined with the organic extension of the CAPRAM 3.0red (Deguillaume et al., 2009) along with a condensed oxidation scheme of malonic acid and succinic acid based on the CAPRAM 3.0red (Deguillaume et al., 2009).

A schematic illustration of current multiphase mechanisms used in this model study and the number of considered processes can be found in Fig. S1. Briefly, the applied aqueous-phase mechanism with 395 reactions is coupled with the RACM-MIM2ext gas-phase mechanism (see Tilgner and Herrmann (2010) for further details) with about 277 reactions. The uptake processes of 42 soluble species are included in the mechanism specified according to the Schwartz approach (Schwartz, 1986) considering Henry's law solubility, gas phase diffusion, and mass accommodation coefficient.

## 2.3 Model simulations

A meteorological scenario has been created for the present simulations, in which an air parcel moves along a predefined trajectory including four cloud passages (two daytime (noon) and two nighttime (midnight) clouds) of about 2 hours within 58 hours of modelling time and a non-cloud deliquesced aerosol period at 90 % relative humidity (RH) in the base case. The applied meteorological scenario is depicted schematically in Fig. 1. Additionally, a sensitivity run considering a lower RH level has been performed at 70 % RH after the second cloud passage to investigate non-ideality effects under lower aerosol liquid water (ALW) conditions. To access the role of a non-ideality treatment for modelling multiphase chemistry in concentrated aqueous aerosols, simulations have been performed both with and without a treatment of non-ideality. Moreover, model simulations were carried out for two different environmental scenarios (urban: anthropogenic polluted case, remote: continental background case) considering summer conditions beginning at 0:00 on the 19[th] of June (45°N). The two scenarios use different initial gas compositions based on (Ervens et al., 2003). Applied physical and chemical aerosol initialisation data of the two environmental scenarios are provided in the supplement (see Table S1/S2). For the simulations, a mono-disperse particle population is assumed, with a dry radius of 0.2 µm. For the sake of clarity, the acronyms "NIDU/NIDR" (Non-IDeal-Urban/Remote) are used for simulations considering the treatment of non-ideality for urban (remote) environmental scenarios and the acronym "IDU/IDR" (IDeal-Urban/Remote) for the ideal solutions for urban (remote) environmental scenarios. An overview on the performed model runs including the acronyms used in this study can be found in Table 1.



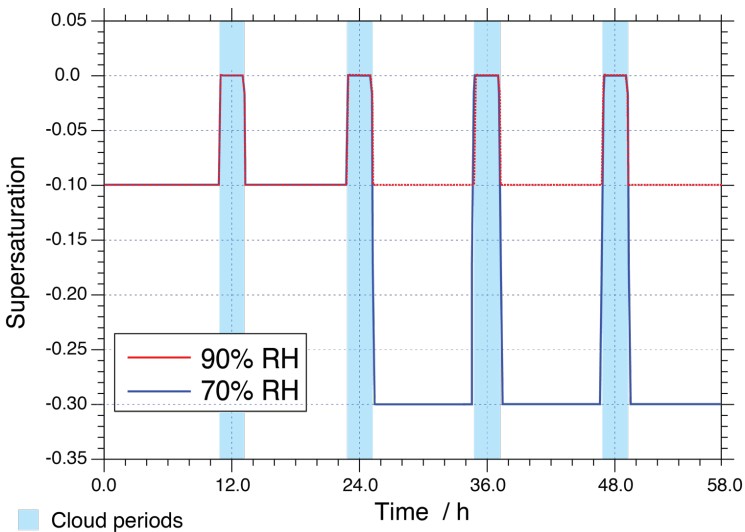

**Figure 1** Depiction of the supersaturation along the trajectories of the two different model scenarios with non-cloud periods at 90 % and 70 % relative humidity.

## 3      Model results and discussions

In this section, SPACCIM-SpactMod simulation results are presented, focusing on (i) the modelled activity coefficients of key compounds under different microphysical conditions, (ii) the impact of a non-ideality treatment on particle acidity and ionic strength, and (iii) the non-ideality impacts on the multiphase processing of key inorganic and organic compounds.

### 3.1   Modelled activity coefficients

According to the treatment of activities instead of concentrations in the NID runs, the chemical fluxes calculated by the model are affected by the predicted activity coefficients. A comprehensive knowledge of the predicted activity coefficients is, therefore, indispensable in order to understand potential differences between model runs with and without a non-ideality treatment. Thus, the predicted activity coefficients are discussed in the beginning for the most important inorganic and organic compounds, and later subsections focus on the non-ideal solution effects on multiphase chemical processing.

First model test applications using SPACCIM-SpactMod in the companion publication (see Rusumdar et al. (2016)) have already briefly addressed the modelled activity coefficients of a few selected compounds. However, the current paper aims at a more comprehensive investigation of the predicted activity coefficients. Moreover, for the sake of clarity, predicted activity coefficients of key inorganic ions and organic compounds are presented separately in the next two subsections.





### 3.1.1   Inorganic ions

The predicted activity coefficients of key inorganic ions and water under deliquesced particle conditions (at 29 hours of modelling time) are tabulated in Table 2 for urban and remote environmental conditions for the two different RH cases of 70 % and 90 % (70 %-NIDU/90 %-NIDR), respectively (see Fig. 1 for details). Additionally, the respective ionic strength and

the water activity of the highly concentrated solutions are given. As enumerated in Table 2, the predicted activity coefficients are presented separately for inorganic anions and cations. Table 2 shows that activity coefficients of inorganic ions are often less than unity mainly due to long-range electrostatic forces in highly concentrated solutions. However, in some cases, activity coefficients of inorganic ions exceed unity, particularly under lower humidity conditions. Furthermore, Table 2 shows that the 70 %-NIDU/70 %-NIDR cases are characterised by higher ionic strengths (21.5 M$^{-1}$/25.8 M$^{-1}$) compared to the 90 %-

NIDU/90 %-NIDR cases (5.2 M$^{-1}$/8.5 M$^{-1}$).

In detail, singly charged anions show activity coefficients within the range of 0.43-1.03 (0.29–0.66), whereas for the double anion $SO_4^{2-}$ a substantially lower value of 0.02 (0.02) is predicted in the base case (90 %-NIDU (90 %-NIDR)). A similar behaviour including a strong deviation of the predicted activity coefficients from the ion charge state is also observed for cations. For singly charged cations, the predicted activity coefficients are in the range of 0.25-0.38 (0.13-0.30), whereas for

doubly and triply charged cations substantially lower values are calculated, with 0.07-0.19 (0.02-0.21) and 0.001 (0.001) under urban (remote) conditions (90 %-NIDU/90 %-NIDR case), respectively.

A comparison of the predicted ion activity coefficients with a small number of values applied in former chemical model studies (Mao et al., 2013; Guo et al., 2014) shows reasonable agreements for singly charged anions and cations, but larger differences for doubly and triply charged cations (see Table 3). The above-mentioned studies have considered constant activity coefficients

for some aerosol constituents or have applied an activity coefficient of a representative compound, i.e. calculated activity coefficients of a selected ion assuming specific ionic strength, aerosol liquid water (i.e,, RH) etc. conditions. Thus, the studies applied one single time-constant activity coefficient representative for singly, doubly, and triply charged anions and cations, ignoring individual ion interaction properties of single ions and RH dependencies. Table 3 shows that the considered activity coefficients in former model studies lie in the range of the coefficients predicted in the present work. For doubly and triply

charged ions, larger deviations are obtainable. Nevertheless, it can be observed that the activity coefficients span a range of values, and an approximation with a single representative value can introduce errors into models. The low value of 0.01 for triply charged ions applied by Mao et al. (2013) is caused by an incorrect implementation of the lower limit estimate by Millero and Woosley (2009). The correct value should be 0.13, which is also much higher than the value predicted in the present study. This deviation is caused by the missing middle-range interaction parameters of iron and manganese in the current model

implementation. In contrast to these two transition metal ions, the $Cu^{2+}$ cation shows a more complex behaviour when lowering RH and increasing ionic strength. The predicted $Cu^{2+}$ cation activity coefficient at 90 % RH is about 0.19 (0.21) in the urban (remote) case, which is similar to the activity coefficient of $Fe^{2+}$ and $Mn^{2+}$. However, at lower RH conditions, the predicted $Cu^{2+}$ cation activity coefficient increases to about 3.7 and 1.75 under urban and remote conditions, respectively. This increase



cannot be observed for $Fe^{2+}$ and $Mn^{2+}$. The behaviour of activity coefficients of metal ions to be lower than unity with increasing ionic strength down to a certain minimum followed by an increase to values partly above unity with further increasing ionic strength is known for many metal ions (see Millero and Woosley (2009) for details). However, due to the missing middle-range interaction parameters of iron and manganese ions in the current model, only the ion-ion interactions

are considered, leading to an activity coefficient below unity only. The known behaviour with further increasing ionic strength can only be obtained for $Cu^{2+}$ and $Mg^{2+}$. This limitation needs to be kept in mind because of the potential impacts on the multiphase chemistry discussed later in Sect. 3.3.

The comparison of the two sensitivity cases (70 %-NIDU/NIDR) reveals that the predicted activity coefficients show a considerable variation between the different RH cases (see Table 2). The predicted activity coefficients do not show a linear

dependency on relative humidity (RH) and water activity coefficient/ionic strength. The water activity coefficient increases by about 0.04 under urban conditions, and decreases by about 0.01 while decreasing RH under remote conditions. In contrast, it can be seen that the activity coefficients of both anions and cations are often lowered while decreasing RH (see Table 2). The highest percentage decreases can be observed for the triply charged ions $Fe^{3+}$ and $Mn^{3+}$, which are lowered by a factor of about 3/2 in the 70 %-NIDU/NIDR cases compared to the base cases (90 %-NIDU/NIDR). However, absolute differences are

very low for both of these triply charged ions under urban (remote) conditions. Still, it has to be kept in mind that, due to the missing middle-range interaction parameters of iron and manganese ions in the current model, only the ion-ion interactions are considered, which leads to an overestimation of this effect. Interestingly, the smallest percentage decrease/increase in the activity coefficients of the considered singly charged ions has been found for $OH^-$ and $H^+$, showing changes of only 12 % (8 %) and 7 % (13 %) between the 70 %-NIDU(NIDR) and 90 %-NIDU(NIDR) runs. Furthermore, a comparison of these

model runs shows that the activity coefficients of singly charged halogen ions (except $I^-$) are higher at lower RH conditions.

In total, activity coefficients of inorganic ions are often considerably lower than unity and notable differences exist between urban and remote aerosols. Thus, it can be expected that the multiphase processing of inorganic ions is mostly decreased in comparison to model runs assuming ideal conditions. However, for some ions, the activity coefficients tend to increase again under more concentrated conditions (lower RH conditions). Consequently, a RH decrease leads to both lowered and increased

activity coefficients, implying an even more reduced or partly increased chemical processing where inorganic ions are involved. Finally, the obtained differences in water activity will subsequently affect microphysical parameters, i.e. radius and saturation ratio. Finally, it needs to be mentioned that due to the missing salt formation processes in the present chemical mechanism, the observed values, in particular those for lower RH conditions, might be slightly biased. Consequently, the present studies have not treated aerosol conditions below 70 %, where insoluble salt and complex formation processes will

play an increasing role.



### 3.1.2 Organic compounds

Similar to inorganic ions, the predicted activity coefficient values of key organic compounds are tabulated in Table 4 under deliquesced particle conditions (at 29 hours of modelling time) for both urban and remote environmental conditions and the two different RH cases. As can be seen, the predicted activity coefficients of organic compounds show a quite uneven pattern overall with values both below and above unity, depending on the functional subgroups of the corresponding compound and the modelled environmental as well as RH conditions. The predicted activity coefficients were observed to be quite variable even though they are chemically very similar and differ only within the functional subgroups (e.g., $CH_2$, OH, and/or COOH).

For organic compounds with one or more alcohol functionalities (i.e. without other functionalities), the calculated activity coefficients are usually larger than unity under remote conditions. Exceptions are methanol and hydrated formaldehyde, which partly shows values slightly below unity. Organic compounds such as hydrated glycolaldehyde and glyoxal, including more than three OH functionalities, show activity coefficients of about 1.1-3.1 and 1.1-2.9, respectively, in the two RH cases (90 %-NIDU/70 %-NIDU). The higher activity coefficients are calculated in the 70 % RH runs. Table 4 shows that activity coefficients for mono alcohols and gem-diols are mostly lower under urban conditions than under remote conditions. Moreover, activity coefficients for mono alcohols and gem-diols generally increase while decreasing RH under urban and remote conditions, respectively.

For carbonyl compounds, the behaviour of the activity coefficients of the different compounds is more even. For aldehydes (without other substituents, except HCHO) and substituted carbonyl compounds (see Table 4), activity coefficients are modelled above unity, whereas activity coefficients increase with an increasing carbon chain length. Similar to alcohols, activity coefficients modelled for aldehydes increase with a decreasing RH.

Activity coefficients of monocarboxylic acids, which are often present in deliquesced particles, are less than unity for dissociated acid anions and in most cases around unity for undissociated acids. Only carboxylic acids without further substituents and a longer non-polar carbon chain (e.g. pyruvic and butyric acid) show higher activity coefficients, particularly with decreasing RH. Non-unity activity coefficients for aerosol components have previously been inferred from other measurements (Jang et al., 1997; Mukherji et al., 1997). Interestingly, the predicted activity coefficients of smaller protonated and deprotonated mono acids are partly quite close. The activity coefficients of dissociated acids are expectedly lower but not significantly so. In Table 4, it can further be seen that the predicted activity coefficients of organic acid anions are largely in the same range as those of inorganic anions. A similar behaviour can be observed for dicarboxylic acids anions and their corresponding iron complexes (see Table 3). Mono anions are characterised by higher activity coefficients than dianions, whereas undissociated diacids show activity coefficients partly above unity, particularly when they do not contain many additional substituents. The values of predicted activity coefficients (see Table 3) that are less than unity are somehow unexpected, especially because this will lead to an increased partitioning of these compounds in the particle phase. As argued by Cappa et al. (2008), the vapour pressures of individual components show strong, identity-dependent deviations from ideality





(i.e., Raoult's Law), with the vapour pressures of the smaller, more volatile compounds decreasing significantly in the mixtures. In addition, in their experimental investigations, they found that the activity coefficients for some of the organic compounds are also less than unity, as the present model results show. Furthermore, based on the obtained numerical values, it can be expected that the non-ideal behaviour of these compounds can modify their gas-particle partitioning.

Overall, the present study shows that activity coefficients of organic compounds are quite variable and compound-specific. Thus, a quite uneven pattern is predicted, with values both below and above unity, leading to both increased and decreased multiphase processing and partitioning of organic compounds due to the treatment of non-ideal conditions. Resulting chemical effects of a non-ideal treatment on key organic subsystems are discussed in detail in the following section.

### 3.1.3  Other compounds

For the sake of completeness, it should be noted that the non-ideality of some non-electrolyte compounds, including key tropospheric oxidants such as OH, $NO_3$, $O_3$ and $H_2O_2$, are not yet considered in SPACCIM-SpactMod (see Rusumdar et al. (2016) for details). They are considered with a constant activity coefficient equal to unity. This is a slight limitation of the current implementation which needs to be addressed in the future. For highly concentrated salt solutions, it is known that non-electrolyte compounds can also influence the thermodynamic equilibrium, although the corresponding interactions will

generally be much weaker than charge interactions. Accordingly, inorganic non-electrolytes can be treated in principle in the same way as organic non-electrolytes in the current SPACCIM-SpactMod implementation (Walther, 1997; Rusumdar et al., 2016). However, this approach requires specific short-range interaction parameters. When such parameters are available the SPACCIM-SpactMod implementation will be improved. Advantageously, such an implementation would allow a detailed calculation of the activity coefficients considering the dependence on the electrostatic interactions of the treated non-ideal

solution, i.e. an activity coefficients calculation as a function of the salt composition. However, for many of the considered non-electrolyte compounds, e.g. radical compounds; such short-range interaction parameter are not available yet. Thus, other simpler estimation methods need to be applied. From the literature (see Marini (2007) and references therein), it is well-known that logarithm of the activity coefficient of neutral solutes is a linear function of the effective ionic strength and the Setchenow (also: Setschenow, Sechenov) coefficient (Setchenow, 1892). This relation is typically applied to calculate the activity

coefficients of neutral solutes and determine their salting-in/-out behaviour when Setchenow coefficient and effective ionic strength is known (Oelkers and Helgeson, 1991). Unfortunately, Setchenow parameters are unknown for many chemical compounds. In this case, available empirical or theoretical prediction methods should be applied to calculate the Setchenow parameters (see Johnson (2010); Yu and Yu (2013) and references therein).

Moreover, it should be mentioned that also radical anions such as $SO_4^-$ and $Cl_2^-$ are not yet treated by the current SPACCIM-

SpactMod and so their activity coefficients are set to unity. In the future, radical anions such as $SO_4^-$ might be treated similarly to their comparable non-radical mono anion ($HSO_4^-$) in case of missing interaction parameters. Finally, all of the above-mentioned model improvements will be part of upcoming SPACCIM-SpactMod studies.





### 3.2 Particle acidity and ionic strength (I)

#### 3.2.1 Particle acidity

Particle acidity and ionic strength (I) are important factors for physico-chemical multiphase chemistry. Thus, the evolution of acidity and ionic strength throughout the simulation was investigated for cloud and deliquesced particle conditions. The $H^+$

activity was initialised by means of a charge balance in SPACCIM. Afterwards, the time evolution of pH is computed dynamically throughout the simulation time (see Sehili et al. (2005) for details). Both particle acidity and ionic strength can be affected by changes in chemical processing and microphysical conditions. Mainly, microphysical parameters such as the ALW content obstruct pH and ionic strength. The modelled evolution of pH and ionic strength of the 90 %-IDU/90 %-NIDU and 70 %-IDU/70 %-NIDU model run throughout the whole simulation time is shown in Fig. 2. The corresponding remote

plot is given in the supplement (see Fig. S2).

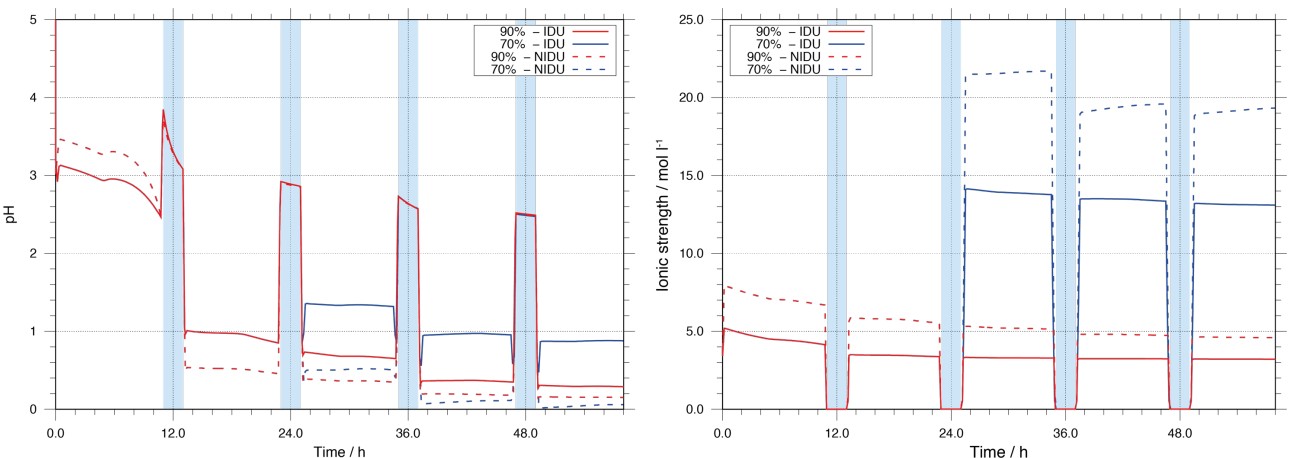

**Figure 2** Modelled pH value (left) and ionic strength (I, right) as a function of simulation time for the urban scenario for the different simulation cases (90 %-IDU/90 %-NIDU and 70 %-IDU/70 %-NIDU).

As shown in Fig. 2, the predicted pH value is initially at around 3 in the performed base case simulations (90 %-IDU/NIDU).

During the first cloud formation period (daytime cloud), the pH increases from about 2.5 under deliquesced particle conditions (at 90 % RH) up to 3.8 under in-cloud conditions ($\approx$ 100 % RH), i.e. the solution becomes less acidic under diluted cloud conditions. During the first daytime cloud, the aqueous-phase oxidation of acid precursors such as $SO_2$ leads to an acidification of the cloud solution and a pH value of about 3.1 at the end of the first daytime cloud. Together with cloud evaporation and the substantial decrease of the ALW, the aerosol pH of the processed aerosol drops down to about 1.0/0.5 (90 %-IDU/NIDU),

which is about 1.5/2.0 lower than the value before cloud processing. Subsequently, the pH slightly decreases further in both cases (90 %-IDU/90 %-NIDU), and the solution becomes more acidic during the aqueous deliquesced particle periods on the first day. Afterwards, the value of pH slightly decreases during the following cloud and deliquesced periods. The simulation





results show substantially lower pH values for the NIDU cases. The predicted pH at the end of the urban case simulation is about 0.15 and 0.3 for the 90 %-NIDU and 90 %-IDU model runs, respectively.

A comparison of the two different RH cases (see Fig. 2) shows larger differences in the predicted pH between the ideal and non-ideal model run for the lower RH case. The predicted pHs at the end of the 70 %-NIDU and 70 %-IDU model runs are about 0.05 and 0.9, respectively. Therefore, the difference of the predicted pHs in the ideal and non-ideal cases is about 0.85 higher between the 70 % RH cases. This tendency of increasing deviations from the ideal solution with decreasing relative humidity has been also observed in experimental and model studies, e.g. Chameides and Stelson (1992); Fridlind and Jacobson (2000); von Glasow and Sander (2001). Nevertheless, experimental investigation of pH in particles is rather difficult, since the ALW contents are usually too small for direct pH measurements (Craig et al., 2018). However, model studies that varied relative humidity were performed mainly for marine environmental conditions (see Fridlind and Jacobson (2000)). Similar studies for remote and urban environmental conditions to compare the present results to are still scarce. Chameides and Stelson (1992) observed a decrease in sea salt aerosol pH when decreasing relative humidity in box model simulations. von Glasow and Sander (2001) argued that the results and the explanation given by Chameides and Stelson (1992) were shown to be insufficient by means of effects of activity coefficients, since microphysical variables also have a certain influence on particle acidity (see von Glasow and Sander (2001)). Moreover, Fridlind and Jacobson (2000) applied the equilibrium model EQUISOLV II in order to analyse the pH of sea salt aerosol for the data obtained through the Aerosol Characterisation Experiment (ACE1) campaign. Their results show that the aqueous-phase aerosol particle pH is less acidic in decreasing relative humidity. Although these results explained the behaviour of the particle pH of marine aerosol particles, similar results were achieved in the sensitivity studies for all the simulations on urban environmental conditions using SPACCIM-SpactMod. During the last simulation period, the 70 %-NIDU case shows slightly lower pH values than the 90 %-NIDU case. In the review of Herrmann et al. (2015), the compiled pH data for measured urban aerosols shows values between -2 and 4. Thus, the predicted pH values of urban conditions of the present study fit into this data range. From the examination of the impact of the non-ideality treatment on pH, it can finally be concluded that the resulting effects on occurring multiphase chemistry (e.g. due to the impact on dissociation equilibriums) should be higher with decreasing relative humidity and, vice versa, that the non-ideality treatment leads to differences in multiphase chemistry, which has feedbacks on acidity.

### 3.2.2 Ionic strength

Besides particle acidity, ionic strength (I) is a key parameter in influencing multiphase chemistry in highly concentrated aqueous solutions. In Fig. 2, the evolution of ionic strength is illustrated for the simulations 90 %-IDU/90 %-NIDU and 70 %-IDU/70 %-NIDU, respectively, throughout the whole simulation time. The corresponding plot for the remote simulations is given in the supplement (see Fig. S2). Figure 2 depicts that the ionic strength predicted for the 90 %-IDU case is rather stable, with values from 5 at the beginning to around 3 mol $l^{-1}$ throughout the simulation time during deliquesced aerosol periods. Just during the well-diluted cloud periods, ionic strength drops down to about $2 \cdot 10^{-3}$-$8 \cdot 10^{-3}$ mol $l^{-1}$. Due to secondary aerosol mass



formation (e.g. via in-cloud sulfur(VI) production), the ionic strength of the in-cloud solution decreases slightly from cloud to cloud. Interestingly, the ionic strength of the aerosol solution stays rather constant throughout the simulation. This behaviour can be explained by two issues. Firstly, connected to the formed soluble secondary aerosol mass, the predicted ALW increase, i.e. the water fraction stays almost constant throughout the simulation time. This compensation mechanism leads to less

increased molarities of important ions such as sulfate. Secondly, ionic strength is buffered in aerosol solutions by the acidity effect. The acidification, due to chemical processing (e.g. in-cloud sulfur(VI) formation), affects the dissociations of acids and, thus, the ratio between singly and doubly charged forms. In the case of sulfur(VI), a shift towards the mono anion form ($HSO_4^-$) is found throughout the modelling time. This shift lowers ionic strength because of the reduced charge number. Overall, both issues buffer an ionic strength increase under deliquesced aerosol conditions. Compared to the ideal base case (90%-IDU), the

90 %-NIDU case shows slightly higher ionic strength levels (approximately 2-3 mol l$^{-1}$ higher during the non-cloud periods). The higher ionic strength values result partly from higher sulfur(VI) concentrations formed in the 90 %-NIDU case compared to the 90 %-IDU case.

In contrast, for the 70 %-IDU/70 %-NIDU cases, the ionic strength increases to about 14/22 mol l$^{-1}$ after the second cloud passage when the air parcel is under lower relative humidity conditions (70 % RH). Whereas just a small difference in ionic

strength is observed for two simulations at 90 % RH, a more distinct difference of about 6-8 mol l$^{-1}$ is observed between the 70 %-IDU and 70 %-NIDU cases. The ideal case (70 %-IDU) always shows somewhat lower ionic strength values than the 70 %-NIDU one. From the present simulations, it can be determined that the ambient RH conditions and the ALW are most likely key impact factors for the ionic strength of the aerosol solution. Finally, the predicted ionic strengths for urban aerosols are in agreement with the data range of literature values (7-45 mol l$^{-1}$) compiled in the review of Herrmann et al. (2015).

**3.3   Multiphase processing of key inorganic compounds**

In this section, the impact of a non-ideality treatment on key inorganic compounds is discussed. As key inorganic chemical compounds, such as sulfur(VI) and transition metals, are present in ionic form, their chemistry can be strongly affected by non-ideal solution effects. Furthermore, this leads to affected oxidant levels that are discussed secondly in this section.

**3.3.1   Sulfur chemistry**

Since S(VI) is one of the main aerosol components substantially formed in the atmospheric aqueous phase, the effects of the non-ideality treatment on S(VI) formation has to be investigated. S(VI) is formed through aqueous-phase oxidation of sulfur dioxide ($SO_2$) through several oxidants (see Tilgner et al. (2013) for details). Figure 3 shows the modelled aqueous-phase S(VI) concentration in µg m$^{-3}$$_{(air)}$ as a function of the modelling time for the urban scenario with and without consideration of non-ideality. As shown in Fig. 3, S(VI) is mainly effectively produced in cloud droplets. Furthermore, the production is higher

in daytime clouds compared to nighttime clouds. Additionally, it can be seen that there is a small difference in the formed S(VI) of the ideal and non-ideal simulations (90 %-IDU/90 %-NIDU). At the end of the simulation, the concentration in the





90 %-NIDU case is about 12 µg m$^{-3}$$_{(air)}$ higher than in the 90 %-IDU case (approximately 20 % more S(VI) production). This result reveals that the treatment of non-ideality plays a minor role in predicting the multiphase processing of S(VI) under cloudy conditions, but might be important under hazy conditions. The observed findings can be explained by a stronger SO$_2$ oxidation in the NIDU cases, where higher H$_2$O$_2$ concentrations are modelled, leading to higher S(IV) oxidations in daytime

aerosol conditions. However, it should be mentioned that the number of available aerosol particles under real haze conditions in strongly polluted areas can be much higher than in the present model study and, thus, the available uptake interface cloud could be larger there. Consequently, S(VI) formation would be less restricted by the uptake of SO$_2$ into the acidic aerosol phase. Due to quite acidic aerosol solution conditions (see Sect. 3.2.1) and a low aerosol surface/ALW volume, the uptake of the weak acid SO$_2$ is limited. Therefore, the aqueous-phase formation of S(VI) is more restricted compared to cloud droplets

in the present simulations, but would be higher under hazy conditions (higher aerosol loadings and ALWs). From the present study, it can be concluded that for simulating S(VI) formation in polluted continental regions, the cloud phase still represents the most important formation medium. However, considering non-ideal solution effects in the chemistry model, higher S(VI) formation fluxes under deliquesced aerosol conditions are modelled. This finding implicates the possibility of a stronger contribution under strongly polluted hazy conditions, which should be investigated in upcoming studies.

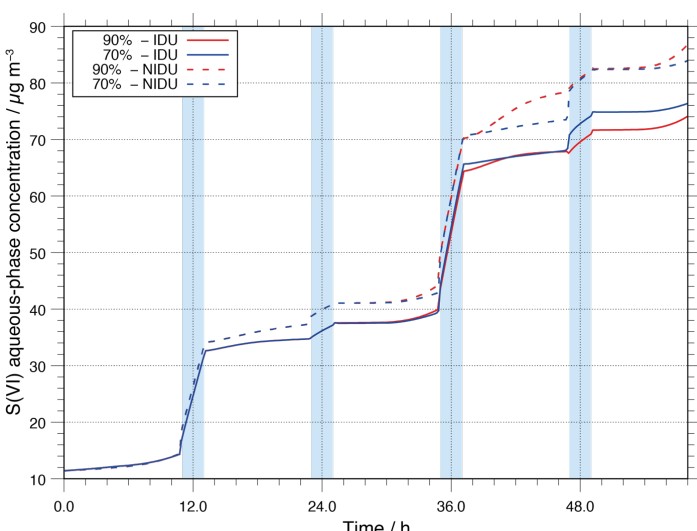

**Figure 3** Modelled sulfur(VI) aqueous-phase concentration in µg m$^{-3}$ throughout the modelling time for the different urban simulation cases (90 %-IDU/90 %-NIDU and 70 %-IDU/70 %-NIDU).

### 3.3.2 Processing of Fe(II)

The time-dependent chemical processing of iron in clouds and deliquesced particles under non-ideal conditions is a quite

important process, because iron speciation and redox-cycling is responsible for several chemical interactions (Deguillaume et al., 2005; Tilgner et al., 2013). Deguillaume et al. (2005) argued in their review that there is still a large uncertainty of aqueous-





phase TMI (transition metal ion) chemistry, since iron speciation is an indicator for atmospheric oxidation and reduction as well as reactivity of aqueous-phase radical chemistry. However, uncertainty remains mainly because of the restricted knowledge about the aqueous particle phase processing of TMIs, such as iron. Tilgner et al. (2013) have shown that the chemical processing of iron in deliquesced particles strongly affects other important chemical subsystems, such as HOx

chemistry. Therefore, the present study is aimed at reducing the uncertainties of multiphase processing of Fe(II) by treating aqueous-phase aerosol chemistry as non-ideal. In the following, only modelled results were presented for the urban environmental scenario (IDU/NIDU) due to the higher concentration levels and influence of iron under these conditions.

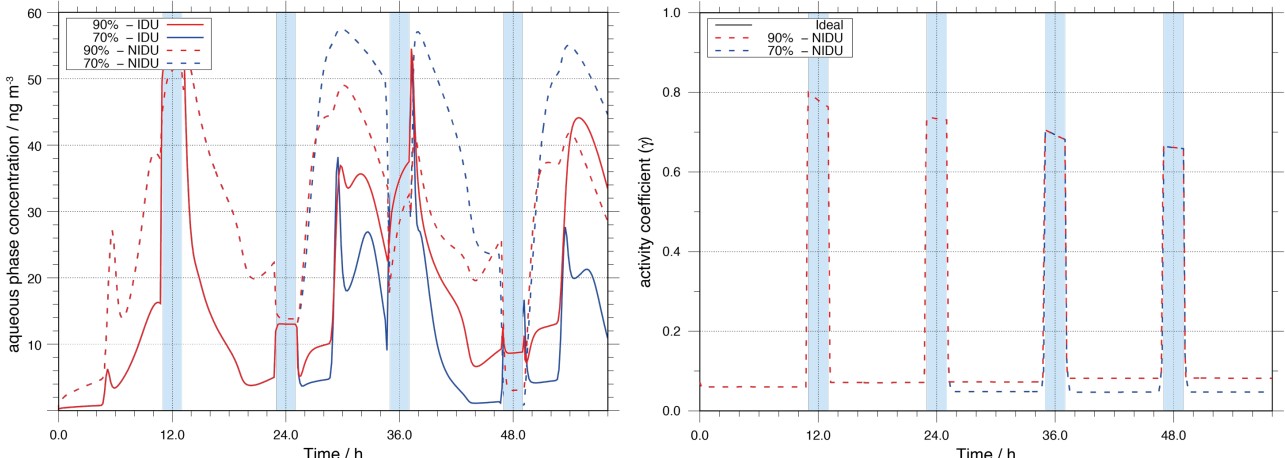

**Figure 4** Modelled Fe(II) aqueous-phase concentration in ng m$^{-3}$ throughout the modelling time (left) and corresponding time evolution of
activity coefficient (right) for the different urban simulation cases (90 %-IDU/90 %-NIDU and 70 %-IDU/70 %-NIDU).

Figure 4 illustrates the aqueous-phase concentration of Fe(II) throughout the simulation time along with the temporal evolution of corresponding predicted activity coefficients in the different urban simulations. As can be seen in Fig. 4, the aqueous-phase concentration is on average a factor of 2.9 higher throughout the whole simulation time for the 90 %-NIDU simulation compared to the 90 %-IDU model run. Especially under deliquesced particle phase conditions, the concentrations of Fe(II) are

substantially higher, indicating a lower oxidation of Fe(II) to Fe(III) in the non-ideal case. The activity coefficient values are less than unity for the whole modelling time (below 0.1 in non-cloud periods), so the non-ideality treatment leads to a reduced multiphase processing of Fe(II). Moreover, Fig. 4 shows that the activity coefficient values tend to have slightly lower values and higher aerosol concentrations of Fe(II) in the 70 %-NIDU case.

A comparison of the modelled total chemical sink and source fluxes of both urban base case model runs (90 %-IDU/90 %-

NIDU) are presented in Fig. 5. There, the modelled chemical sink and source fluxes of Fe(II) in the aqueous phase plotted for the second model of the 90 %-IDU vs. 90 %-NIDU case can be seen. The corresponding plot for the remote cases is presented in the supplement (see Fig. S3). Figure 5 shows a characteristic daytime profile of the redox-cycling fluxes, which is interrupted by cloud periods. The comparison reveals both reduced formation and oxidation fluxes for the 90 %-NIDU case





compared to the 90 %-IDU simulation. This finding shows that the Fe(II) oxidation is less efficient when considering non-ideality. Whereas the chemical fluxes are quite similar under cloud conditions of the 90 %-IDU/90 %-NIDU cases, the chemical fluxes under deliquesced aerosol conditions are substantially lowered. Throughout the simulation time, the fluxes of the 90 %-NIDU case are approximately a factor of 2.8 lower compared to the 90 %-IDU case. Figure 5 reveals that $Fe^{2+}$

5    interacts mainly with other ions of the aerosol solution. The reaction of $Fe^{3+}$ and $Cu^+$ is, for example, the main source of $Fe^{2+}$ under deliquesced aerosol conditions. Due to the calculated activity coefficient of ions (see Table 1), which are predicted as less than unity, the calculated reaction fluxes are lowered under consideration of non-ideality. Hence, the contribution of this reaction during the overall processing of $Fe^{2+}$ is decreased in aqueous particles. As a result, reduced formation fluxes, including the fluxes of the most important sink processes, such as the Fenton reaction of $Fe^{2+}$ and $H_2O_2$, are modelled. This leads to a

10   reduced processing of $HO_x/HO_y$ (see the following two subsections for further details).

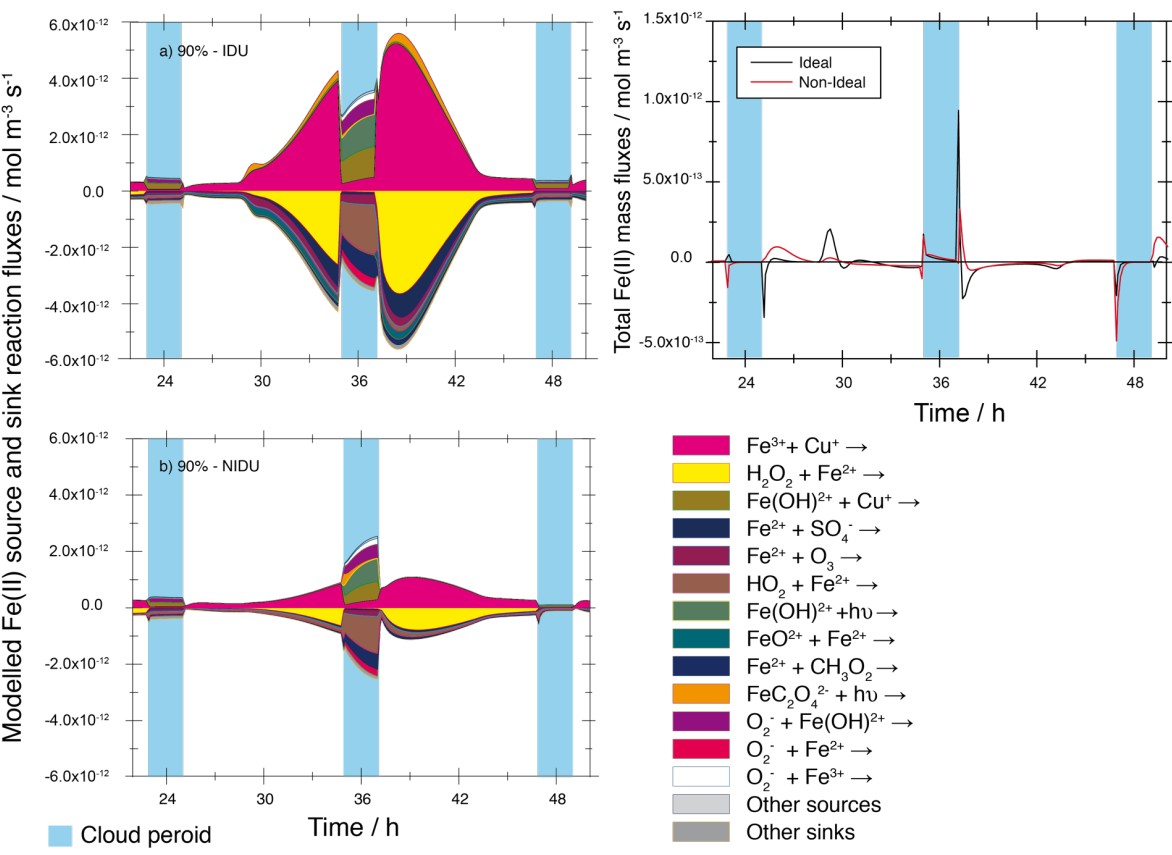

**Figure 5** Modelled chemical sink and source mass fluxes of Fe(II) in the aqueous phase in mol m$^{-3}$ s$^{-1}$ for the second day of the urban cases (90 %-IDU/90 %-NIDU). a) ideal solutions (90 %-IDU), b) non-ideal solutions (90 %-NIDU), c) corresponding total fluxes (sum of source and sink fluxes). Only sinks and sources with a contribution larger than ±3 % are presented.



To summarise, the model simulations implicate that the multiphase processing of Fe(II) processing aqueous particles is significantly affected by the treatment of non-ideality and the effect depends strongly on the ALW conditions. By treating aqueous-phase chemistry as non-ideal solutions, considerably smaller reaction fluxes of Fe(II) have been observed in deliquesced particles compared to ideal solution runs. The impact of this lowered processing on other chemical subsystems, e.g. the changed $H_2O_2$ processing in the aerosol phase, is discussed in the following sections.

### 3.3.3 Non-ideality effect on the $H_2O_2$ budget

Because of the substantially reduced Fe(II) cycling, the resulting effects on the $H_2O_2$ budget have been studied. The time concentration profiles of gaseous and aqueous $H_2O_2$ are plotted in Fig. 6 for both the 90 %-IDU/90 %-NIDU and the 70 %-IDU/70 %-NIDU cases. Figure 6 reveals that due to the consideration of a non-ideality, the predicted $H_2O_2$ concentrations are significantly affected. Figure 6 shows that the predicted gaseous and aqueous concentrations of $H_2O_2$ are higher for the NIDU simulation cases. The comparison of the two base cases (90 %-IDU/90 %-NIDU) shows that the modelled aqueous concentrations of $H_2O_2$ are on average a factor of 3.1 larger during the non-cloud periods in the 90 %-NIDU case. A similar pattern can be observed for the gas-phase concentrations of $H_2O_2$. This finding is interesting because it demonstrates that the treatment of non-ideality has a potential impact on the multiphase (gas and aqueous phase) budget of $H_2O_2$. Using an ideal solution treatment, the gaseous budget of $H_2O_2$ would be underpredicted and less $H_2O_2$ would be available for other processes such as aqueous-phase S(VI) oxidation. The higher $H_2O_2$ concentrations can be explained by chemical fluxes.

The analysis of the chemical sink and source mass fluxes of aqueous $H_2O_2$ reveals higher aqueous-phase formation fluxes (without considering phase-transfer as a source) during the non-cloud periods in the 90 %-NIDU case. The uptake of $H_2O_2$ from the gas phase is an important source of aqueous $H_2O_2$ in the 90 %-IDU case with a contribution of about 48 %. However, the overall uptake flux of $H_2O_2$ from the gas phase is about 22 % lower in the non-ideal simulation (90 %-NIDU). This fact is consistent with the 23 % larger aqueous-phase formation fluxes in the 90 %-NIDU simulation. Besides the lower uptake flux from the gas phase, flux analysis shows an increased contribution of the reaction of $HO_2$ with $Cu^+$ to aqueous $H_2O_2$ formation. The formation flux of $H_2O_2$ by reaction of $HO_2$ with $Cu^+$ is about 32 % larger in the 90 %-NIDU case than in the ideal simulation, and is the most important source with about 53 %. Moreover, the analysis of the chemical sink fluxes of aqueous-phase $H_2O_2$ shows that the most important $H_2O_2$ sink, the reaction with $HSO_3^-$, contributes more in the 90 %-NIDU case (40 % higher fluxes). That is the reason for the higher sulfate formation in the non-ideal case. On the other hand, the overall flux of the other key sinks of $H_2O_2$, the Fenton reaction with $Fe^{2+}$, is about 70 % lower in the 90 %-NIDU case than in the 90 %-IDU one. The reaction flux analysis implies that the contribution of reaction pathways containing doubly or triply charged ions (e.g. the Fenton reaction of $Fe^{2+}$) is reduced and, in contrast, the contribution of reaction pathways containing singly charged ions (e.g. the reaction of $HO_2$ with $Cu^+$ and $HSO_3^-$ oxidation) increases when non-ideality is considered in the simulation. Overall, it can be concluded that the treatment of non-ideality generally leads to higher multiphase $H_2O_2$ concentrations and to changes





in the fluxes of different reaction pathways. Finally, similar results are modelled for the remote cases (see Fig. S4 in the supplement) but with smaller differences than in the urban case.

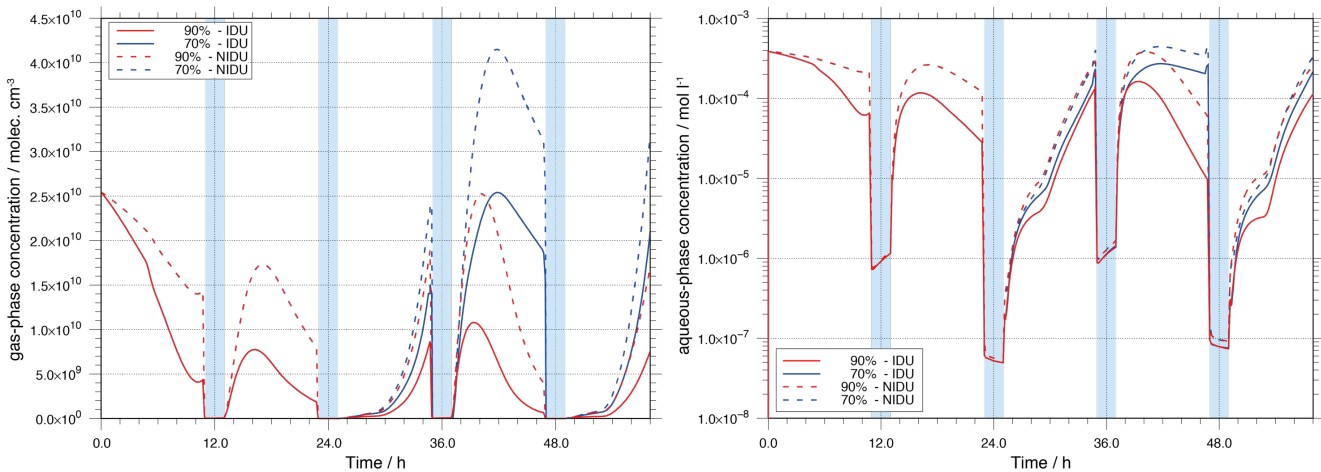

**Figure 6** Modelled gas- and aqueous-phase concentrations of $H_2O_2$ throughout the simulation time for the different urban simulation cases
(90 %-IDU/90 %-NIDU and 70 %-IDU/70 %-NIDU).

## 3.4  Aqueous-phase processing of OH radicals

*Aqueous-phase OH concentrations*

In Fig. 7, the aqueous-phase concentrations of OH in $mol\,l^{-1}$ are plotted for the urban simulations at 90 % RH (90 %-IDU/90 %-NIDU) and 70 % RH (70 %-IDU/70 %-NIDU). The corresponding plots for the remote scenario are presented in the
supplement (see Fig. S5). Due to the strong dependency on both microphysical conditions (e.g. aerosol/cloud water content) and different chemical sink/source fluxes, the aqueous-phase OH radical concentrations show a diurnal profile interrupted by cloud periods. The modelled OH concentrations in the deliquesced particles are higher than the concentrations under in-cloud conditions. Comparing the concentration levels in the deliquesced particles before and after a cloud passage, Fig.7 shows a reduction after cloud evaporation. This reflects effective in-cloud oxidations, which affects OH formation pathways after cloud
passages. The reduction after cloud evaporation is more distinct in the non-ideal cases.

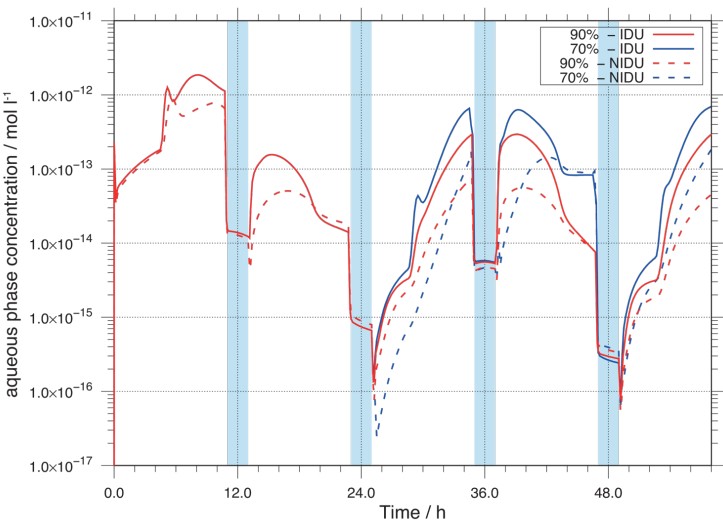

**Figure 7** Modelled OH aqueous-phase concentrations in mol l$^{-1}$ throughout the simulation time for the different urban simulation cases (90 %-IDU/90 %-NIDU and 70 %-IDU/70 %-NIDU).

A comparison of the ideal and non-ideal base case simulations (90 %-IDU/90 %-NIDU) shows substantially higher OH concentrations for the ideal case (90 %-IDU). The ideal base model run (90 %-IDU) is characterised by higher concentrations, on average a factor of 4 higher, than the concentrations in the 90 %-NIDU case under non-cloud conditions. At the end of the 90 %-IDU/90 %-NIDU simulations, deliquesced particle-phase concentrations of about $3 \cdot 10^{-13}/4.5 \cdot 10^{-14}$ are modelled, showing a factor of about 7 higher OH levels under aerosol conditions. On the other hand, the cloud concentrations in both cases are almost similar. This behaviour can be explained by the importance of different sources under deliquesced aerosol and cloud conditions. The in-situ sources are strongly affected by the non-ideality treatment under non-cloud conditions, whereas its impact under cloud conditions is minor.

Interestingly, OH concentrations of the 70 %-NIDU simulation are partly higher than the ones modelled in the 90 %-NIDU case. This change in the concentration pattern can be explained by changes in the reaction fluxes (see below). Moreover, the 70 %-NIDU case also shows higher concentrations during the first half of the second night period than the 90 %-NIDU case. The modelled concentration is almost one order of magnitude higher in the 70 %-NIDU case before the second nighttime cloud ($\approx 47$ h), indicating higher OH formation fluxes in this model case. This also enables higher OH oxidations during this night period (see below for details). At the end of the simulation, the daytime aerosol concentrations of the 70 %-NIDU case are about a factor of 4 higher than in the 90 %-NIDU case, but still lower than in the two ideal simulations.

*Aqueous-phase OH formation and oxidation fluxes*

The significant differences obtained in the ideal and non-ideal concentration patterns suggest that the activity coefficients of the reaction partners of OH have a strong influence on the modelled chemical sinks and sources and, finally, the aqueous-





phase concentrations of OH. Even though the activity coefficients for neutral radicals such as OH are considered as unity in the present model, the influence of non-ideality has been considered for the computation of reaction rates, when the radicals react with other organic/inorganic compounds. As tabulated in Table 2 and Table 4, the activity coefficients of key inorganic and organic OH reaction partners strongly vary. Subsequently, the chemical mass fluxes of sink and source reactions for the

5    OH radical can be increased or decreased depending on individual activity coefficients. The differences in the aqueous-phase reaction fluxes are depicted in Fig. 8.

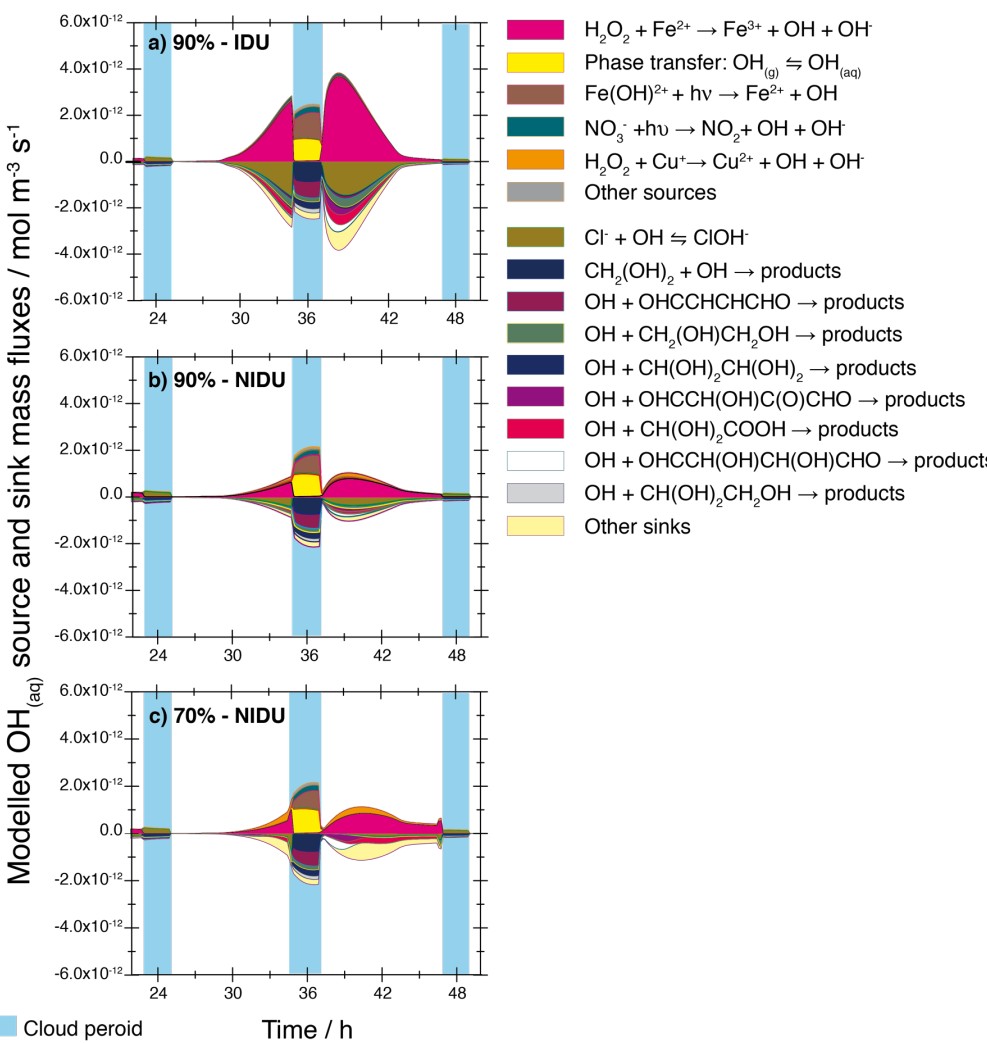

**Figure 8** Modelled chemical sink and source mass fluxes of OH in the aqueous phase in mol m$^{-3}$ s$^{-1}$ for the second day of modelling time for the urban scenario for the simulations 90 %-IDU vs. 90 %-NIDU vs. 70 %-NIDU. a) ideal solutions (90 %-IDU), b) non-ideal solutions

10   (90 %-NIDU), and c) non-ideal solutions (70 %-NIDU).



In detail, Fig. 8 shows the chemical mass fluxes of different OH reaction pathways (sinks/sources) for the simulations 90 %-IDU, 90 %-NIDU, and 70 %-NIDU for the urban scenario throughout the second model day. Figure 8 shows only minor differences in the chemical fluxes under cloud conditions where non-ideality effects are minor and are therefore not further discussed. Larger differences can be obtained during the deliquesced aerosol periods. As illustrated in Fig. 8, both the total source and sink fluxes are lowered in the deliquesced particle phase when non-ideality is considered. Further, the peak in the source and sink fluxes is slightly delayed towards the later afternoon for the 70/90 %-NIDU simulations compared to the 90 %-IDU one. Due to the treatment of non-ideality, the total OH formation flux of the 90 %-NIDU case is reduced by about 52 % throughout the whole simulation time compared to the 90 %-IDU case, and by a factor of $\approx 4$ in the afternoon peak on the second day (see Fig. 8). Interestingly, a flux comparison of the 70 %-NIDU and 90 %-NIDU simulations reveals a slightly higher afternoon peak on the second day and markedly higher fluxes in the first half of the second nighttime period in the 70 %-NIDU case. This finding implies higher OH oxidation fluxes in aqueous aerosols at nighttime in the more concentrated case (70 %-NIDU). Overall, the total OH flux in the deliquesced particle periods of the 70 %-NIDU case is about 8 % higher than in the 90 %-NIDU one. This finding implies that the chemical processing in deliquesced particles might be more favourable under more concentrated solutions rather than higher humidity aerosol conditions.

A more detailed look at the specific reaction fluxes reveals that, in all simulation cases, the OH production under aqueous particle conditions is strongly related to the Fenton reaction of $Fe^{2+}$. This means that in-situ OH production strongly depends on both $Fe^{2+}$ and $H_2O_2$ concentrations. This finding is in agreement with former CAPRAM mechanism studies (see Tilgner et al. (2013)). Additionally, a comparison of the formation flux pattern of the two non-ideal cases reveals that the Fenton-like reaction of copper contributes noticeably more to the aqueous-phase formation of OH in deliquesced particles. This additional source leads to the slightly higher OH formation fluxes in the 70 %-NIDU case than in the 90 %-NIDU one. The stronger contribution of this OH source can be explained by both the substantially higher activity coefficient of $Cu^+$ under 70 % RH conditions (see Table 2), leading to higher chemical fluxes, and the larger $H_2O_2$ budget in the 70 %-NIDU case. The latter additionally causes higher formation fluxes in the 70 %-NIDU case until the second nighttime cloud. After its evaporation, the formation fluxes are lowered as a consequence of the substantially lowered $H_2O_2$ budget due to efficient in-cloud sulfur oxidation.

As shown in Fig. 8, the OH sinks are also affected in the deliquesced particle periods due to the treatment of non-ideality. Figure 8 shows a change in the sink flux pattern between the 90 %-IDU and 70/90 %-NIDU cases, respectively. In aqueous particles, the relative OH radical loss by reactions with organic compounds and the gaseous OH uptake is increased and, on the other hand, the fraction of $Cl^-$ interaction with OH is lowered under 90 %-NIDU conditions compared to the ideal base case. Throughout the whole simulation, the $Cl^-$ interactions with OH contribute about 36 %, 26 %, and 14 % to the OH sink fluxes during the non-cloud periods in the 90 %-IDU, 90 %-NIDU, and 70 %-NIDU cases, respectively. Accordingly, the relative contributions of OH oxidation reactions of organic compounds, particularly substituted carbonyl compounds and undissociated organic acids, often increase in the non-ideal cases. Moreover, a comparison of the two non-ideal simulations,





90 %-NIDU and 70 %-NIDU, shows differences (see Fig. 8) in the sink reactions under deliquesced particle conditions. In the 70 %-NIDU case, OH reactions of substituted carbonyl compounds and undissociated organic acids are more important than in the 90 %-NIDU one. For example, the contribution of the OH oxidation of pyruvic acid/glycolic acid (undissociated) to the total sink fluxes (over 58 h) is increased from 0.7 %/2.8 %, 0.8 %/3.6 % to 2.6 %/4.7 % in the 90 %-IDU, 90 %-NIDU, and 70 %-NIDU cases, respectively. The raised contribution of substituted organic compounds can be partly explained by the higher activity coefficients of such compounds under more concentrated conditions (see Table 4 and Sect. 3.1.2). The raised degradations of some organic undissociated acids and other substituted organic compounds leads to an affected concentration pattern of those compounds (see the following subsection for details).

Overall, the present model investigations reveal that the processing of OH radicals (formation and loss processes) in aqueous particles is reduced, considering the treatment of non-ideality of concentrated solutions, mainly because of the lowered in-situ formation of OH. Furthermore, the treatment of non-ideality leads to differences in the chemical sink and source contributions of different chemical pathways. However, the results here strongly depend on the non-ideality effects of transition metal ions and microphysical conditions (ALW conditions). As discussed above, missing middle-range interaction parameters of iron and manganese ions in the current model lead to activity coefficients below unity only for iron and manganese ions. This limitation may lead to a strong under-prediction of the Fenton reaction fluxes and thus to undervalued OH formations and aqueous-phase oxidation fluxes.

### 3.5  Multiphase processing of organic compounds

The aqueous-phase chemical processing of organic compounds is expected to not only be limited to in-cloud conditions but also to proceed in aqueous particles with significant chemical fluxes (Tilgner and Herrmann, 2010; Tilgner et al., 2013). In contrast to former studies that assumed aqueous-phase chemistry as ideal solutions, the present SPACCIM-SpactMod study is aimed at investigating the effects of a non-ideality treatment on the organic processing of important organic $C_2$ and $C_3$ oxidation pathways.

### 3.5.1  $C_2$ organic chemistry

As shown in many former model studies (Ervens et al., 2003; Herrmann et al., 2005; Tilgner and Herrmann, 2010; Ervens et al., 2011), in-cloud oxidations of $C_2$ carbonyl compounds such as glycolaldehyde and glyoxal lead to formation-substituted organic acids (e.g. glyoxylic and glycolic acid), which can be further oxidised into oxalic acid, contributing to aqSOA under both cloud and aerosol conditions. However, available studies within the literature have not yet considered the effects of non-ideality and therefore have to be deemed incomplete.

Figure 9 shows the modelled aqueous-phase mass concentrations of oxalic acid and its precursors, glyoxylic and glycolic acid, along with corresponding activity coefficients vs. simulated time under urban environmental conditions for the 90 %-





IDU/90 %-NIDU and 70 %-IDU/70 %-NIDU simulations, respectively. For the sake of clarity, the sums of dissociated and undissociated aqueous-phase concentrations of the carboxylic acids are plotted in Fig. 9.

**Figure 9** Modelled aqueous-phase concentrations in ng m$^{-3}_{(air)}$ and corresponding activity coefficients for the important C$_2$ oxidation products, (i) glycolic acid (top), (ii) glyoxylic acid (centre), and (iii) oxalic acid (down). The plotted concentrations represent the sum of the dissociated and undissociated forms of the acids.





As shown in Table 4 and Fig. 9, the activity coefficients of dissociated and undissociated forms of the above-mentioned acids are different. The predicted activity coefficients of the undissociated acids are substantially higher than those of the dissociated forms. These differences in the activity coefficients of dissociated and undissociated forms of the organic acids, together with the different oxidant budget, affect the chemical processing of those acids in deliquesced particles along with the molarity of the compound under different RH conditions.

The predicted concentration-time profiles presented in Fig. 9 show that the oxalic acid precursors, glyoxylic and glycolic acid, are effectively produced under cloud conditions. Furthermore, it can be seen that their degradation proceeds almost in deliquesced particles. Both findings are in agreement with former studies (see e.g., Tilgner et al. (2013)).

*Glycolic and glyoxylic acid*

For glycolic and glyoxylic acid, the modelled in-cloud productions in the 90 %-NIDU and 90 %-IDU cases are similar. However, due to the incorporation of activity coefficients, lower degradations in the 90 %-NIDU than in the 90 %-IDU case are modelled in aqueous aerosols. Furthermore, the production in the aqueous particle phase is also not similar. Throughout the whole simulation period, the deviation increases with the simulation time for 90 %-IDU and 90 %-NIDU.

The modelled concentrations of glycolic acid in the 90 %-NIDU case are substantially higher at about 135 ng m$^{-3}$ at the end of the simulation time compared to the 90 %-IDU case at about 40 ng m$^{-3}$. A comparison of the two non-ideal cases (90 %-NIDU/70 %-NIDU) reveals a faster and longer ongoing degradation of glycolic acid in the deliquesced particle phase under the lower humidity conditions of the 70 %-NIDU case. At the end of the simulation, the modelled glycolic acid is, surprisingly, closer to the modelled 90 %-IDU case than to the 90 %-NIDU one at about 50 ng m$^{-3}$. The modelled glycolic acid concentration pattern reveals substantially higher degradation levels in the afternoon of the second day and the first half of the second night period (37 h-47 h) until nighttime cloud formation. Therefore, the final glycolic acid concentration is much lower in the 70 %-NIDU case than in the 90 %-NIDU one. Due to the considered non-ideality treatment, both formation and degradation fluxes are decreased in the non-ideal 90 %-NIDU case. The formation and degradation fluxes of glycolic acid in the ideal case are 28 % and 74 % larger, respectively, throughout the whole simulation time. Moreover, the comparison of the time-resolved overall sink fluxes shows 13 % and 71 % higher formation fluxes in the 90 %-IDU case under in-cloud and aqueous aerosol conditions, respectively. Furthermore, the sink fluxes under aerosol conditions are about 2 times larger than in the non-ideal base case. Consequently, the larger sink fluxes in the ideal case (90%-IDU) reflect that the resulting glycolic acid concentrations are higher in the 90 %-NIDU case than in the 90 %-IDU one. The lower oxidation of glycolic acid in the 90 %-NIDU case, particularly in the afternoon, also leads to a shift in the contributions of oxidants to the overall oxidation flux. Whereas the overall oxidation flux of glycolic acid via OH is lowered by 50 % due to the strongly lowered OH budget, the NO$_3$ oxidation flux of glycolic acid is increased instead, particularly during the nighttime cloud phase. During that phase, glycolic acid is more present in its dissociated form, electron transfer reactions are favoured more, and the concentration budget is higher because of the lowered daytime decay. A similar picture can be seen for glyoxylic acid. The modelled urban





concentration profiles of glyoxylic acid show the same trend as glycolic acid. Thus, a discussion of the glyoxylic acid concentration pattern and fluxes is omitted in the present study for the sake of clarity.

*Oxalic acid*

Besides glyoxylic acid and glycolic acid, Fig. 9 also depicts the concentrations of its key oxidation product, oxalic acid.

From the reduced degradation flux of the substituted mono carboxylic acids, one could expect lower oxalic acid concentrations under non-ideal conditions (90 %-NIDU). However, the lowered formation fluxes of oxalic acid do not lead directly to lower predicted concentrations of oxalic acid. The treatment of non-ideality significantly affects the chemical sinks of oxalic acid and the predicted degradation fluxes are lower in cases treating non-ideality effects. Thus, the resulting oxalic acid levels at the end of the simulation are substantially higher in the 90 %-NIDU and particularly in the 70 %-NIDU case. In the 90 %-

NIDU case, the final concentration is a factor of about 1.8 higher than in the 90 %-IDU case. The difference in the predicted oxalic acid mass is even higher in the 70 % cases (70 %-NIDU/70 %-IDU). There, the predicted oxalic acid concentration at the end of the simulation is a factor of 10 higher compared to the model without non-ideality treatment (70 %-IDU). A similar tendency with higher predicted oxalic acid concentrations in the non-ideal cases is also observed in the remote simulations, but at lower concentration levels (see Fig. S6 in the supplement).

The higher Fe(II) concentrations, the reduced chemical processing of iron, and the modelled activity coefficients of the different oxalic acid anions and iron-oxalate complex ions in the simulations with non-ideality treatment leads to a smaller complexation of the diacid. This results in substantially lower fluxes of the photochemical decompositions of the iron-oxalate complexes and thus to higher oxalic acid concentrations in both the 90 %-NIDU and 70 %-NIDU cases. A comparison of the modelled reaction fluxes (90 %-IDU/90 %-NIDU, see Fig. 10) shows, firstly, a distinct reduction of the formation fluxes and,

secondly, a drastic change in the decomposition flux pattern.



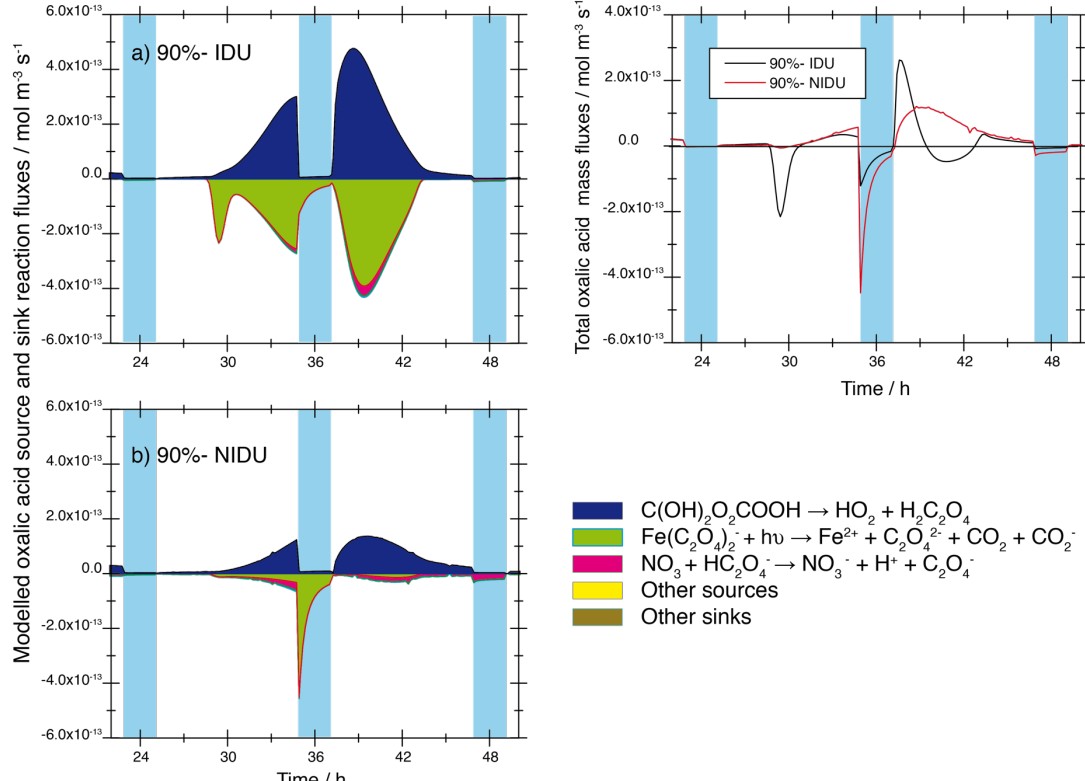

**Figure 10** Modelled chemical sink and source mass fluxes of oxalic acid/oxalate in the aqueous phase in mol m⁻³ s⁻¹ for the second day of modelling time for the urban scenario for the simulations 90 %-IDU vs. 90 %-NIDU. a) Ideal solutions (90 %-IDU), b) non-ideal solutions (90 %-NIDU), c) corresponding total fluxes. Only sinks and sources with a contribution larger than ±3 % are presented.

As discussed in Tilgner and Herrmann (2010), the formation of oxalic acid mostly takes place in the aqueous phase of deliquesced particles. As shown in Fig. 10, oxalic acid is effectively produced by the oxidation of glyoxylic acid, predominantly during the day, especially in deliquesced particles. On the other hand, oxalic acid is also substantially decomposed via the fast photo-catalytic decay of iron-oxalate complexes. Consequently, quite low oxalic acid concentrations are predicted in the ideal base case (90 %-IDU). A comparison of the modelled flux pattern (90 %-IDU/90 %-NIDU) in Fig. 10 reveals that the computed overall formation and degradation reaction mass fluxes were decreased by factor of 2.5 and 2.9, respectively, for the 90 %-NIDU simulation compared to the 90 %-IDU one. This stronger reduction leads to higher oxalic acid concentrations. Moreover, the sink flux pattern shows high values during cloud periods in the non-ideal case (90 %-NIDU) and only small photolytic decay fluxes during non-cloud periods. This change in sink pattern is caused by the non-ideality treatment that lowers the formation of iron-oxalate-complexes. The interaction of the doubly charged oxalate and the triply charged $Fe^{3+}$ ion is reduced due to the modelled activity coefficients of such ions that strongly depend on the charge number. Due to the high charge numbers of the complexing agents (high LR interactions), the iron complex formation is less





efficient than in the ideal case. Consequently, efficient photolysis is hampered and less oxalate is decomposed. In total, the present studies demonstrate that the treatment of non-ideality can significantly affect multiphase oxidation and the resulting concentration budget of important $C_2$ carboxylic acids. The non-ideality treatment also enables more realistic predictions of high oxalate concentrations as observed in field campaigns under highly polluted conditions (van Pinxteren et al., 2014; Kawamura and Bikkina, 2016; Zhu et al., 2018).

### 3.5.2   $C_3$ organic chemistry

As presented in former CAPRAM studies (e.g. Tilgner and Herrmann (2010)), methylglyoxal is effectively oxidised under cloud conditions, leading to the formation of substituted $C_3$ carboxylic acids, which are expected to be further oxidised in aqueous aerosols. In Fig. 11, the aqueous-phase concentrations of the main $C_3$ oxidation products, such as pyruvic acid, 3-oxo pyruvic acid, and finally keto malonic acid, along with corresponding activity coefficients are plotted throughout the simulation time for all urban scenarios. The corresponding plots for the remote scenario are presented in the supplement (see Fig. S7).

Figure 11 shows that the first stage oxidation product, pyruvic acid, is effectively produced under both day- and nighttime cloud conditions. Noticeably, urban nighttime clouds reveal a higher production of pyruvic acid than corresponding daytime clouds. This finding is in agreement with former studies (Tilgner and Herrmann, 2010; Tilgner et al., 2013), showing that in-cloud $NO_3$ radical oxidation represents an important sink for methylglyoxal and, consequently, a source for pyruvic acid under urban conditions.

Figure 11 illustrates that the activity coefficients of formed oxidation products are below unity for their dissociated forms in the 90 %-NIDU and 70 %-NIDU simulations under deliquesced aerosol conditions. However, their undissociated forms partly show activity coefficients above unity. Consequently, the non-ideality treatment can affect their further chemical processing differently in addition to the affected oxidant budget. In detail, undissociated pyruvic acid is characterised by activity coefficients above unity (about 1.5 in the 90 %-NIDU case) and its dissociated form by values of about 0.4 (90 %-NIDU case). Thus, non-ideality effects can increase the chemical processing of pyruvic acid and decrease the chemical processing of pyruvate. Under 70 %-NIDU conditions, pyruvic acid and pyruvate are characterised by even more different activity coefficients of about 3.0 and 0.3, respectively. Therefore, even higher differences between the ideal and the non-ideal treatment can occur.





**Figure 11** Modelled aqueous-phase concentrations in ng m$^{-3}_{(air)}$ and corresponding activity coefficients for the important C$_3$ oxidation products, (i) pyruvic acid (top), (ii) 3-oxo-pyruvic acid (centre), and (iii) keto malonic acid (down). The plotted concentrations represent the sum of dissociated and undissociated forms of the carboxylic acids.

5    Figure 11 shows that the in-cloud formations of pyruvic acid and keto malonic acid are similar in the 90 %-NIDU and 90 %-IDU cases. Differences between predicted concentration curves arise mainly during deliquesced aerosol periods. For pyruvic





acid, the ideal simulation is characterised by a faster degradation during aqueous aerosol conditions. In contrast, the 90 %-NIDU case shows a much lower decrease. This lower oxidation flux leads to the higher concentrations of pyruvic acid in the 90 %-NIDU case than in the 90 %-IDU case. Interestingly, Fig. 11 illustrates that the decrease is lower in the 90 %-NIDU case but higher under 70 %-NIDU conditions. A comparison of the 90 %-NIDU vs. 70 %-NIDU concentration profiles shows a

stronger oxidation in the more concentrated solution case in the afternoon and evening of the second model day. Therefore, lower pyruvic acid concentrations and, consequently, higher 3-oxo pyruvic acid concentrations can be found in the 70 %-NIDU simulation during this period. The reasons for these higher oxidation levels in the aerosol phase of the 70 %-NIDU case are (i) the higher OH budget in the 70 %-NIDU case (see sSect. 3.3) and (ii) the substantially higher activity coefficients of pyruvic acid under 70 % RH conditions, which leads to faster degradation. A reaction flux investigation of OH radicals has

already revealed the higher degradation fluxes of pyruvic acid under lower humidity conditions (see Sect. 3.3). An analysis of the reaction fluxes of pyruvic acid reveals degradation fluxes that are a factor of 2.7 higher in the 70 %-NIDU than in the 90 %-NIDU case during the aerosol period of 37-47 h.

Subsequently, further oxidation of 3-oxo pyruvic acid mainly occurs during cloud periods and leads to the formation of keto malonic acid (see Fig. 12). Keto malonic acid is the final $C_3$ oxidation product of this oxidation chain, which is formed

particularly under cloud and daytime deliquesced particle conditions. Due to the non-ideality treatment, the predicted concentrations of keto malonic acid are lower when compared to the ideal cases. The predicted concentration of keto malonic acid at the end of the 90 %-NIDU simulation is a factor of 2 lower.

From Fig. 12, it can be seen that the generally lowered oxidation of the precursors leads to a smaller production of keto malonic acid. On the other hand, the lowered oxidation of keto malonic acid compensates for the lowered formation fluxes. In detail,

both the total keto malonic acid formation and sink fluxes are approximately 50 % lower in the 90 %-NIDU compared to the 90 %-IDU case throughout the whole simulation time. This interplay causes the relatively small differences in the predicted concentrations modelled between the ideal and non-ideal cases. Furthermore, Fig. 12 shows that both the formation patterns change in a way that makes keto malonic acid show less chemical aerosol phase fluxes and more distinct in-cloud fluxes. This means that the non-ideality treatment could both decrease (e.g. in 90 %-NIDU) and increase (e.g. in 70 %-NIDU) the chemical

turnovers in the aerosol phase.



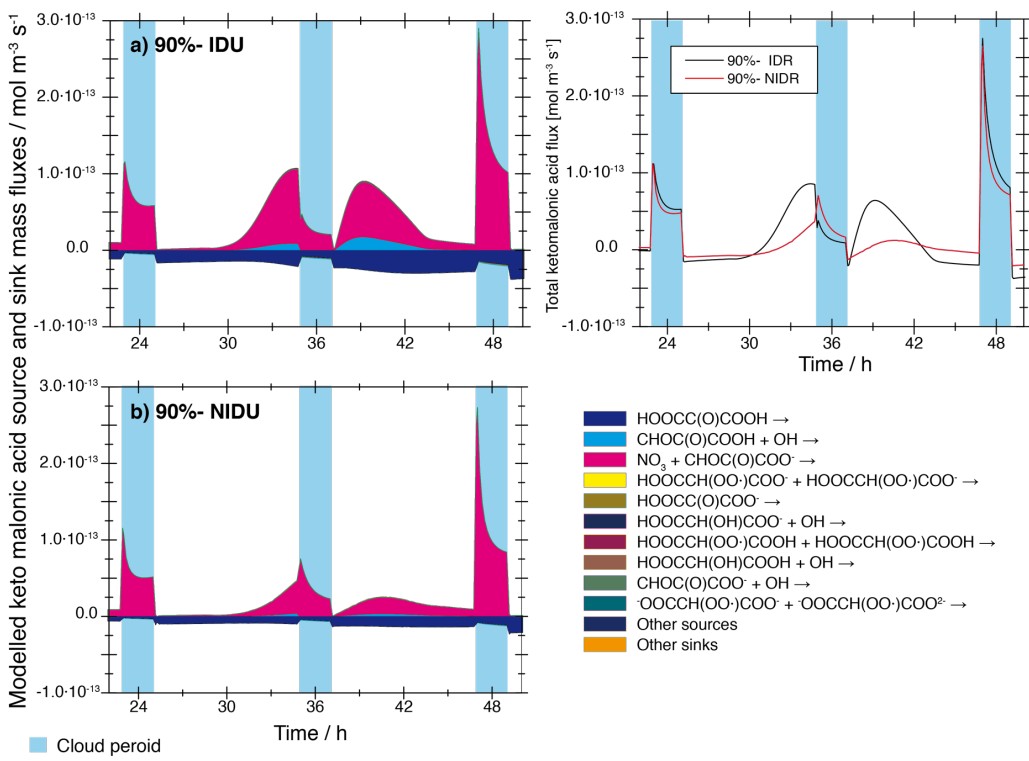

**Figure 12** Modelled chemical sink and source mass fluxes of keto malonic acid in the aqueous phase in mol m$^{-3}$ s$^{-1}$ for the second day of modelling time for the urban scenarios, 90 %-IDU vs. 90 %-NIDU. a) Ideal solutions (90 %-IDU), b) non-ideal solutions (90 %-NIDU)

Overall, the present investigations of organic chemistry have demonstrated that the processing of organic compounds and, subsequently, their concentrations in aqueous particles can be affected substantially by considering the treatment of non-ideality of concentrated solutions. The effects are mainly caused by the changed oxidants budget and the different activity behaviours of the different organic compounds. Moreover, the non-ideality treatment leads to substantial changes in the chemical sink and source pattern compared to the ideal simulations.

## 4 Conclusions

For the first time, detailed simulations with the advanced parcel model SPACCIM-SpactMod have been carried out for the different microphysical conditions with and without a treatment of non-ideality for aqueous-phase aerosol chemistry. Special emphasis was put on the different chemical subsystems, including key inorganic compounds, radical and non-radical oxidants, and organic compounds, in order to examine the effects of a non-ideality solution treatment. The simulation results highlight that a treatment of activities instead of concentrations strongly affects the chemical multiphase processing in deliquesced particles.





The investigations of the predicted activity coefficients have revealed substantial differences between different charged and uncharged aerosol compounds. For inorganic ions, activity coefficients are often considerably lower than unity under 90 % RH conditions and strongly decrease with the increasing charge state of the respective ion because of the impact of long-range electrostatic forces in highly concentrated solutions. In detail, the predicted activity coefficients of singly, doubly, and triply

charged inorganic ions are in the range of 0.25-1.03 (0.13–0.66), 0.02-0.19 (0.02-0.21), and 0.001 (0.001), respectively, under urban(remote) 90 % RH conditions. Interestingly however, the activity coefficients of some inorganic ions exceed unity, particularly under lower humidity conditions when the aerosol solution is even more concentrated. Consequently, an RH decrease can cause both lowered and raised activity coefficients, leading to more reduced or increased chemical processing for pathways involving inorganic ions. However, once more, increasing activity coefficients with values larger than unity

under lower humidity conditions have only been modelled for some metal ions, such as $Cu^{2+}$ and $Mg^{2+}$. Unfortunately, the current model cannot model such an increase for other chemically important transition metal ions, e.g. $Fe^{2+}$ and $Mn^{2+}$. The behaviour of the activity coefficients of metal ions to be lower than unity with increasing ionic strength down to a certain minimum, followed by an increase to values partly above unity with further increasing non-ideality is strongly related to the middle-range interaction forces. However, due to missing middle-range interaction parameters in the current model, e.g. of

iron and manganese ions, only ion-ion interactions are considered, leading to activity coefficients below unity only. This limitation might have potential impacts on the multiphase chemistry in the current model, probably leading to an underestimation of certain chemical processes under lower RH conditions. Therefore, the present model studies implicate that there is a demand for further improvements of the current model implementation by middle-range interaction parameters from future laboratory studies.

The predicted activity coefficients of organic acid anions are in the same range of those of inorganic anions. However, the behaviour of the activity coefficients of uncharged organic compounds is partly different because of their dependence on the nature of intermolecular interaction forces. Thus, the predicted activity coefficients of organic compounds are quite variable and compound-specific. This uneven pattern of organic compounds with values of predicted activity coefficients of both below and above unity implies an increased or decreased aqueous phase processing due to the treatment of solution non-ideality. In

general, the activity coefficients for alcohols, gem-diols, aldehydes, dialdehydes, as well as undissociated mono and dicarboxylic acids with longer carbon chains are observed as larger than unity, whereas smaller carboxylic acids and particularly carboxylate ions are observed as below unity. The activity coefficients strongly depend on functional and $CH_x$ groups contained in organic molecules. Moreover, the present study demonstrates that activity coefficients of many uncharged organic compounds, generally strongly increase while decreasing the RH under both urban and remote conditions.

As a consequence of the non-ideal solution treatment, the present model investigations show that the chemical multiphase processing of inorganic compounds is often strongly affected. The model simulations reveal that the multiphase processing of Fe(II) in aqueous particles is significantly reduced by the treatment of non-ideality and the effect depends strongly on the ALW conditions. The decreased Fe(II) processing has a substantial effect on other chemical subsystems leading to, for example, changed $H_2O_2$ and OH processing in aqueous aerosols.





In the case of $H_2O_2$, the model simulations illustrated that due to the consideration of a non-ideality treatment, the predicted $H_2O_2$ concentrations are significantly higher in both the gas and the aqueous phase. This effect is caused by the significantly lowered degradation flux of $H_2O_2$, mainly the Fenton reaction, and increased formation fluxes, e.g. via the reaction of $HO_2$ with $Cu^+$. On the other hand, higher chemical rates of the sulfur oxidation, $H_2O_2$ reaction with $HSO_3^-$, have been modelled.

Thus, this study implicates that the treatment of non-ideality has a potential impact on $H_2O_2$, resulting in higher multiphase concentrations of this oxidant. This higher oxidant level is, consequently, available for other processes, such as a more aqueous-phase S(VI) oxidation leading to a higher aerosol mass. The present study has shown approximately 20 % more S(VI) production compared to an ideal urban model run.

The smaller importance of the Fenton reaction under aqueous aerosol conditions is shown to also affect other oxidants besides

$H_2O_2$. The present model investigations reveal that the processing of OH radicals (formation and loss processes) in aqueous particles is substantially lowered considering the treatment of non-ideality of concentrated solutions, mainly as a consequence of the reduced in-situ formation of OH by the Fenton reaction. Additionally, the non-ideality treatment produces chemical differences in the sink and source patterns of different chemical pathways. However, this finding strongly depends on the non-ideality effects of transition metal ions and the microphysical conditions (ALW). Therefore, the missing middle-range

interaction parameters of, for example, iron and manganese ions could introduce uncertainties in the current model. This may cause a too strong underprediction of the Fenton reaction and may thus lead to an undervalued OH formation and OH-initiated aqueous-phase oxidation fluxes. Thus, this issue needs further investigations in upcoming model studies.

Due to the affected concentration budget and the rates of oxidants, significant effects on multiphase oxidations and the resulting concentrations of important organic aerosol components have been observed by comparing ideal and non-ideal solution

simulations. Interestingly, the reduced aqueous-phase OH oxidation budget leads to lowered oxidations in deliquesced particles and thus to higher concentration levels of oxidised compounds mainly produced in the cloud phase. On the other hand, the present simulations have demonstrated that production and degradation pathways can be asymmetrically influenced, resulting, for example, in higher oxalate concentrations. Although the oxalate precursors are less further oxidised in deliquesced aerosols and the oxalate formation fluxes are therefore reduced, the simulations with a non-ideality treatment revealed higher oxalate

concentrations because of the even stronger reduced photolytic decays of iron-oxalate complexes. Thus, non-ideal simulations enable more realistic predictions of high oxalate concentrations as observed in field campaigns under highly polluted conditions. Moreover, this also makes artificial disregards of the photolytic decays of iron-oxalate complexes in some other multiphase chemical mechanisms (e.g. Ervens et al. (2004)) unnecessary. Furthermore, the present study implicates that lower humidity conditions, characterised by more concentrated solutions, might promote the formation of higher oxalic acid

concentration levels in aqueous aerosols. In conclusion, the present investigations of organic chemistry have shown that the processing of organic compounds and subsequently their concentrations in aqueous particles can be affected substantially by considering the treatment of non-ideality of concentrated solutions. The effects are mainly caused by the changed oxidants budget and the different activity coefficient behaviour of the different organic compounds. Moreover, the non-ideality treatment leads to substantial changes in the chemical sink and source pattern compared to the ideal simulations. Nevertheless,



it should be kept in mind that the present results are strongly related to the non-ideality effects of transition metal ions, oxidants, and ALW (microphysical) conditions. Therefore, the present results have to be considered as an initial step for further studies with more advanced model versions in the future.

Overall, the model studies have implicated the importance of the consideration of the treatment of non-ideality for tropospheric aerosol constituents for the first time, especially in the deliquesced particle phase. From the current SPACCIM-SpactMod model studies along with detailed reaction flux investigations, it can be concluded that the treatment of non-ideality is highly necessary in multiphase models to gain an understanding of physico-chemical multiphase processing in aqueous aerosols. Likewise, the current model studies reveal the need for further detailed analysis in order to understand the effect of non-ideality and unsolved issues of multiphase processing of aqueous aerosol particles by adopting more complex chemistry with a high number of organic compounds. Further studies also need to be performed for different meteorological conditions, i.e. marine environmental conditions. Additionally, adequate consideration of liquid-liquid separations as well as salt formation (crystallization) in deliquesced aerosol particles is also necessary, which can potentially alter the ionic strength and acidity, and thus influence non-ideality. Moreover, model advancements are necessary by extending the database with new organic and inorganic AIOMFAC interaction parameters (Ganbavale et al., 2015; Gervasi et al., 2019).



*Data availability.* The model initialisation data are provided in the Appendix. Further SPACCIM-SpactMod datasets and model simulation data of this study can be accessed by contacting the corresponding authors (H. Herrmann: herrmann@tropos.de / R. Wolke: wolke@tropos.de).

*Supplement.* The supplement related to this article is available online at:

*Author contributions.* AJR, AT, and RW have designed and performed the SPACCIM-SpactMod model simulations. AJR, AT, RW and HH have analysed model results. AT, RW, AJR and HH have written the paper.

*Competing interests.* The authors declare no conflicts of interest.

*Special issue statement.* This article is part of the special issue "Simulation chambers as tools in atmospheric research (AMT/ACP/GMD inter-journal SI)". It is not associated with a conference.

*Acknowledgements.* Part of this work was performed within the project ACoMa funded by the German Research Foundation (Deutsche Forschungsgemeinschaft, DFG) under the project number TI 925/1-1. This work has received funding from the European Union's Horizon 2020 research and innovation programme through the EUROCHAMP-2020 Infrastructure Activity under grant agreement no 730997.

*Financial support.* This research has been supported by the Deutsche Forschungsgemeinschaft (grant no. TI 925/1-1) and the
European Commission (grant no. EUROCHAMP-2020 (730997)).

*Review statement.*



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



**Tables**

**Table 1.** List of the microphysical model scenarios and their acronyms used in this study.

| Case | Chemistry treatment | Environmental conditions | Acronym |
|---|---|---|---|
| Base case (90% RH) | Ideal | Remote | 90%-IDR |
| | | Urban | 90%-IDU |
| | Non-Ideal | Remote | 90%-NIDR |
| | | Urban | 90%-NIDU |
| | | | |
| 70% RH | Ideal | Remote | 70%-IDR |
| | | Urban | 70%-IDU |
| | Non-Ideal | Remote | 70%-NIDR |
| | | Urban | 70%-NIDU |



**Table 2.** Predicted activity coefficients of selected ions and water, as well as the ionic strength in deliquesced particles for the 90 %/70 %-NIDR and 90 %/70 %-NIDU simulations at 29 hours. Δγ represents the ratio of the predicted activity coefficients of the 90 % case and the 70 % case, respectively.

| Compound | Urban | | | Remote | | |
|---|---|---|---|---|---|---|
| | 90 % | 70% | Δγ | 90 % | 70% | Δγ |
| **Inorganic anions** | | | | | | |
| $SO_4^{2-}$ | 0.02 | 0.02 | 1.02 | 0.02 | 0.02 | 1.01 |
| $HSO_4^-$ | 1.03 | 0.93 | 1.11 | 0.64 | 0.72 | 0.89 |
| $NO_3^-$ | 0.44 | 0.23 | 1.95 | 0.29 | 0.18 | 1.58 |
| $OH^-$ | 0.53 | 0.60 | 0.88 | 0.54 | 0.59 | 0.92 |
| $F^-$ | 0.59 | 0.78 | 0.76 | 0.66 | 0.93 | 0.71 |
| $Cl^-$ | 0.75 | 0.87 | 0.87 | 0.58 | 0.67 | 0.86 |
| $Br^-$ | 0.85 | 1.07 | 0.80 | 0.63 | 0.79 | 0.80 |
| $I^-$ | 0.43 | 0.32 | 1.35 | 0.40 | 0.30 | 1.34 |
| **Inorganic cations** | | | | | | |
| $H^+$ | 0.38 | 0.35 | 1.07 | 0.13 | 0.12 | 1.13 |
| $NH_4^+$ | 0.26 | 0.18 | 1.43 | 0.25 | 0.18 | 1.37 |
| $Na^+$ | 0.37 | 0.31 | 1.20 | 0.30 | 0.25 | 1.17 |
| $K^+$ | 0.25 | 0.16 | 1.54 | 0.26 | 0.20 | 1.29 |
| $Mg^{2+}$ | 0.08 | 0.30 | 0.26 | 0.02 | 0.08 | 0.24 |
| $Fe^{2+}$ | 0.07 | 0.05 | 1.49 | 0.06 | 0.05 | 1.25 |
| $Mn^{2+}$ | 0.07 | 0.05 | 1.49 | 0.06 | 0.05 | 1.25 |
| $Cu^{2+}$ | 0.19 | 3.70 | 0.05 | 0.21 | 1.75 | 0.12 |
| $Fe^{3+}$ | $1.3\cdot10^{-3}$ | $4.6\cdot10^{-4}$ | 2.94 | $9.1\cdot10^{-4}$ | $4.3\cdot10^{-4}$ | 2.14 |
| $Mn^{3+}$ | $1.3\cdot10^{-3}$ | $4.6\cdot10^{-4}$ | 2.94 | $9.1\cdot10^{-4}$ | $4.3\cdot10^{-4}$ | 2.14 |
| Water activity coefficient | 1.02 | 1.06 | 0.97 | 1.05 | 1.04 | 1.00 |
| Water activity | 0.90 | 0.70 | 1.29 | 0.90 | 0.70 | 1.29 |
| **Ionic strength** | 5.21 | 21.53 | 0.24 | 8.46 | 25.77 | 0.33 |





**Table 3:** Comparison of activity coefficients applied in multiphase aerosol chemistry models. The data of the present study is based on Table 1.

| Cations / Anions | Mao et al. (2013) | Guo et al. (2014) | Present study |
|---|---|---|---|
| $X^+$ | 0.6 | 0.8 (based on $Na^+$) | 0.29-1.03 |
| $Y^-$ | 0.6 | 0.6 (based on $HSO_4^{2-}$) | 0.13-0.38 |
| $X^{2+}$ | $\approx 0.067$ (based on $Cu^{2+}$) / | 0.45 (based on $Fe^{2+}$) | 0.02-0.21 |
| $Y^{2-}$ | — | 0.02 (based on $SO_4^{2-}$) | 0.02 |
| $X^{3+}$ | 0.01 | 0.3 (based on $Fe^{3+}$) | 0.001 |





**Table 4.** Predicted activity coefficients of selected organic compounds in deliquesced particles for the 90 %/70 %-NIDR and 90 %/7 0%-NIDU simulations at 29 hours. Δγ represents the ratio of the predicted activity coefficients of the 90 % case and the 70 % case, respectively.

| Species | Urban | | | Remote | | |
|---|---|---|---|---|---|---|
| | 90 % | 70% | Δγ | 90 % | 70% | Δγ |
| *Alcohols* | | | | | | |
| Methanol (CH₃OH) | 0.88 | 1.08 | 0.82 | 0.94 | 1.20 | 0.78 |
| Ethanol (CH₃CH₂OH) | 1.17 | 3.47 | 0.34 | 1.26 | 3.43 | 0.37 |
| *Aldehydes* | | | | | | |
| *Formaldehyde* | | | | | | |
| CH₂OH₂ (hydrated) | 0.87 | 1.02 | 0.85 | 1.06 | 1.48 | 0.71 |
| HCHO | 0.87 | 0.99 | 0.88 | 0.85 | 1.00 | 0.84 |
| *Acetaldehyde* | | | | | | |
| CH₃CHO | 1.23 | 3.78 | 0.33 | 1.36 | 3.54 | 0.38 |
| CH₃CHOH₂ (hydrated) | 1.29 | 4.42 | 0.29 | 1.42 | 4.41 | 0.32 |
| *Propionaldehyde* (CH₃CH₂CHO) | 1.62 | 12.12 | 0.13 | 1.83 | 10.14 | 0.18 |
| *Butyraldehyde* (CH₃CH₂CH₂CHO) | 2.14 | 38.90 | 0.06 | 2.45 | 29.04 | 0.08 |
| *Glycolaldehyde* | | | | | | |
| OHCCH₂OH | 1.21 | 3.57 | 0.34 | 1.54 | 4.39 | 0.35 |
| OH₂CHCH₂OH (hydrated) | 1.13 | 3.10 | 0.36 | 1.61 | 5.28 | 0.30 |
| *Glyoxal* | | | | | | |
| CHOH₂CHOH₂ (hydrated) | 1.11 | 2.93 | 0.38 | 1.82 | 6.56 | 0.28 |
| CHOCHO | 1.27 | 3.89 | 0.33 | 1.67 | 4.54 | 0.37 |
| *Methylglyoxal* | | | | | | |
| CH₃C(O)CH(OH)₂ (hydrated) | 1.75 | 9.39 | 0.19 | 2.60 | 18.00 | 0.14 |
| *Monocarboxylic acids* | | | | | | |
| *Formic acid* | | | | | | |
| HCOOH | 0.85 | 0.47 | 1.8 | 0.75 | 0.39 | 1.92 |
| HCOO⁻ | 0.43 | 0.29 | 1.50 | 0.41 | 0.29 | 1.40 |
| *Acetic acid* | | | | | | |
| CH₃COOH | 0.97 | 1.06 | 0.92 | 0.84 | 0.70 | 1.20 |
| CH₃COO⁻ | 0.43 | 0.29 | 1.46 | 0.40 | 0.29 | 1.36 |
| CH₃CH₂COOH | 1.28 | 3.39 | 0.38 | 1.13 | 2.00 | 0.56 |
| CH₃CH₂CH₂COOH | 1.69 | 10.89 | 0.16 | 1.51 | 5.73 | 0.26 |
| *Glycolic acid* | | | | | | |
| CH₂(OH)COOH | 0.95 | 1.00 | 0.95 | 0.95 | 0.87 | 1.10 |
| CH₂(OH)COO⁻ | 0.43 | 0.29 | 1.46 | 0.40 | 0.29 | 1.36 |
| *Glyoxylic acid* | | | | | | |
| CH(OH)₂COOH | 0.94 | 0.94 | 0.99 | 1.08 | 1.08 | 1.00 |
| CH(OH)₂COO⁻ | 0.43 | 0.29 | 1.46 | 0.40 | 0.29 | 1.36 |
| *Pyruvic acid* | | | | | | |
| CH₃C(O)COOH | 1.48 | 3.03 | 0.49 | 1.54 | 2.96 | 0.52 |
| CH₃C(O)COO⁻ | 0.43 | 0.29 | 1.46 | 0.40 | 0.29 | 1.36 |
| *Dicarboxylic acids* | | | | | | |
| *Oxalic acid* | | | | | | |
| H₂C₂O₄ | 0.79 | 0.30 | 2.59 | 0.64 | 0.18 | 3.61 |





| Species | Urban | | | Remote | | |
|---|---|---|---|---|---|---|
| | **90 %** | **70%** | **Δγ** | **90 %** | **70%** | **Δγ** |
| $C_2O_4^{2-}$ | 0.05 | 0.03 | 1.89 | 0.04 | 0.03 | 1.61 |
| $HC_2O_4^-$ | 0.43 | 0.29 | 1.46 | 0.40 | 0.29 | 1.36 |
| $[Fe(C_2O_4)_2]^-$ | 0.43 | 0.29 | 1.46 | 0.40 | 0.29 | 1.36 |
| $[Fe(C_2O_4)]^+$ | 0.43 | 0.29 | 1.46 | 0.40 | 0.29 | 1.36 |
| $[Fe(C_2O_4)_3]^{3-}$ | $1.3\cdot10^{-3}$ | $4.6\cdot10^{-4}$ | 2.94 | $9.1\cdot10^{-4}$ | $4.3\cdot10^{-4}$ | 2.14 |
| *Malonic acid* | | | | | | |
| $HOOCCH_2COOH$ | 1.04 | 0.98 | 1.07 | 0.86 | 0.51 | 1.69 |
| $HOOCCH_2COO^-$ | 0.43 | 0.29 | 1.47 | 0.41 | 0.29 | 1.39 |
| $^-OOCCH_2COO^-$ | 0.05 | 0.03 | 1.89 | 0.04 | 0.03 | 1.61 |
| *Succinic acid* | | | | | | |
| $C_2H_4COOH_2$ | 1.38 | 3.13 | 0.44 | 1.15 | 1.45 | 0.79 |
| $HOOCC_2H_4COO^-$ | 0.39 | 0.23 | 1.70 | 0.36 | 0.23 | 1.54 |
| $^-OOCCH_2CH_2COO^-$ | 0.05 | 0.03 | 1.89 | 0.04 | 0.03 | 1.61 |
| *Keto malonic acid* | | | | | | |
| $HOOCCOCOOH$ | 0.88 | 0.66 | 1.33 | 0.86 | 0.67 | 1.28 |
| $HOOCCOCOO^-$ | 0.43 | 0.29 | 1.46 | 0.40 | 0.29 | 1.36 |
| $^-OOCCOCOO^-$ | 0.05 | 0.03 | 1.89 | 0.04 | 0.03 | 1.61 |
| *Malic acid* | | | | | | |
| $HOOCCHOHCH_2COOH$ | 1.39 | 4.08 | 0.34 | 1.45 | 3.20 | 0.45 |
| $HOOCCHOHCH_2COO^-$ | 0.43 | 0.29 | 1.46 | 0.40 | 0.29 | 1.36 |
| $^-OOCCHOHCH_2COO^-$ | 0.05 | 0.03 | 1.89 | 0.04 | 0.03 | 1.61 |