# Peer review of "Treatment of non-ideality in the multiphase model SPACCIM-Part2: Impacts on the multiphase chemical processing in deliquesced aerosol particles"

_Atmospheric Chemistry and Physics, 2019_

## Referee Comment (RC1) · Anonymous Referee #1 · 9 Jan 2020

The study presented in this paper aims to quantify the activity coefficients of species dissolved in deliquescent atmospheric particles. The approach is based on the modeling of two continental scenarios (urban, remote) simulating the chemical evolution in various phases (gas – particle - cloud droplets). The study examines the activity coefficients of inorganic (HOx, SOx, H2O2, metals) and organic (C1-C3) species. The effect of the non-ideal behavior of the species on the dynamic evolution of the system is evaluated by comparing the results obtained when the ideal behavior in the aqueous phase is imposed or not. For the scenarios tested, the budget of key species (S(VI),

[Figure]

H2O2, organic acids ...) is substantially modified when the non-ideal behavior of the solution is considered. The results presented are relevant to this journal. The paper is well organized but the writing could likely be more concise. I have a few comments (see below) that the authors might consider for submitting a revised article to ACP.

1. The description of the two scenarios considered in this study is inadequate. The authors refer to an article by Ervens et al. (2003) (page 7, line 24), but I did not find more detailed information in the article cited. Additional information is necessary to characterize the simulations carried out with the box model (initial conditions, emissions, speciation of organic species, deposition, NOx regime, NH3 level, etc.) and to weigh the simulated behavior of the species (e.g. the accumulation of S(VI) presented in fig.3, page 16 and reaching 90 $\mu$g/m3). The details of the simulation conditions could for example be added to the additional material.

2. The paper examines the effect of the non-ideality of the solution on the condensed phase reactivity but does not discuss its effect on the gas/particle partitioning, as suggested for example page 12, line 4. No result of the non-ideal behavior is presented on the simulated aerosol mass concentrations, their compositions (e.g. inorganic vs organic), their liquid water content ... Adding a section presenting the effects on the overall aerosol properties (mass concentration, composition, water content in particular) would be useful to evaluate the sensitivity of this non-ideal behavior in models dealing with aerosols formation.

3. The iron activity decreases notably when the non-ideal behavior of the species is considered. The authors show that this decrease in activity is one of the main causes explaining the simulated effects. However, the authors state that the model ignores the "middle range interaction" for Fe and Mg ions, leading to uncertainties about the activity coefficients calculated for these species. This limitation is reminded very regularly in the paper, including in the conclusion of the article (e.g. line 15, page 34). Although clearly mentioned, this uncertainty considerably weakens the conclusions of this study. I think that a test would be necessary to examine the sensitivity of the simulations to

uncertainties about the activity of iron. An educated guess could perhaps be made to estimate the effect of the middle range interactions, e.g. using another TMI as a proxy.

4. Minor comments: The term "flux" is used systematically by the authors to define the rate of processes. I think that using "flux" is inappropriate here and should be replaced by "rate" everywhere in the text and in figures 5, 8, 10 and 12.

The writing of the chemical formula are often difficult to read for organic molecules. For example in Table 4, OH2CHCH2OH would be more easy to read as CH(OH)2CH2OH to represent hydrated glycolaldehyde. Similarly, C2H4COOH2 would be more easy to read as C(O)OHCH2CH2C(O)OH to represent succinic acid, etc . . . Furthermore, the writing is sometime misleading, like in figure 8 for OHCCHCHCHO (which I guess is for butenedial CHOCH=CHCHO) or, in figure 10, C(OH)2O2COOH which I cannot identify (a typo?) but understand to be glyoxylic acid from the text. I suggest to use only "conventional" formula in the text, figures and tables.

Page 2, line 20: the word "including" in the context of the sentence is unclear. Change to "leading to"?

Page3, line 16: the paper use both ALWC and ALW. Duplication of 2 similar acronyms could be avoided (ALWC is used only 2 times in paper).

Page 10, line 26-27: One of the 2 "finally" should be removed.

Page 11, line 22: pyruvic acid is not a simple carboxylic acid (i.e. without another functional group), as suggested in the context of the sentence.

Page 12, line 18-20. The sentence seems useless in the context of the paragraph and could be removed.

Page 13, line 8. I don't understand the meaning of the word "obstruct" in the context of the sentence.

Page 14, line 5-6: the sentence is obvious in the context of the paragraph and could

be removed.

Page 15, line 14-16: The sentence is obvious in the context of the paragraph and could be removed.

Page 19, line 1: the second processing in "processing of Fe(II) processing aqueous" could be removed.

Page 19, line 17-31: The reading of this paragraph is tedious. A figure similar to 5, 8, 10 or 12 could be useful.

Page 21, line 7: units are missing for OH concentration. Number of significant figures should be consistent for the 2 numbers.

Page 24, line 25: I don't understand "formation-substituted" in the sentence.

Page 26, line 10-32: The reading of this paragraph is tedious. A figure similar to 5, 8, 10 or 12 could be useful.

Page 31, line 8: Typo in "sSect."

Page 31, line 14: it is stated that keto malonic acid is the final C3 oxidation product. Figure 12 (page 32) show however a loss process which is not discussed in the text (and not "explained" from the legend in the figure).

Page 32, figure 12: Only the first 3 processes (dark blue, light blue and pink) can be discerned in the figure. The other processes could be merged in either the "other source" or "other sinks" boxes.

---

## Referee Comment (RC2) · Anonymous Referee #2 · 22 Jan 2020

The authors present a sensitivity study of the impact of considering nonideal solution effects on predictions of aqueous aerosol chemistry using the SPACCIM-SpactMod model. The study is well-conceived, and the results highlight the potential significance of nonideal solution effects in multiphase atmospheric chemistry, something that is generally ignored in atmospheric models due in part to computational expense but also because of a lack of data. I think the manuscript will be suitable for publication in ACP after a few clarifications.

- The aqueous chemical kinetic data in CAPRAM, which the multiphase chemical kinetic simulations in this study were based on, for the most part was not originally measured/reported as a function of activity but rather of concentration. The measurements that form the basis of that data were also mostly performed under much more dilute and closer to ideal conditions than the aqueous aerosol conditions considered here. How accurate is it to use those rate laws but plug in the activities calculated here - which are in many cases very different from the activities where the rates were originally measured both because of the high concentrations of atmospheric aerosols but also because of the nonideality? Is it even possible to evaluate this?

- I agree with the other referee's comment that the uncertainty in the MR parameterizations for Fe and Mn are a significant weakness, especially given the likely importance of those two ions in multiphase atmospheric chemistry. Could the authors estimate those parameters, perhaps from available lab data, to at least constrain the effect?

- Considerable uncertainty exists in the measured and reported rate data for much TMI chemistry, especially when it comes to ionic strength and pH dependence. Can the authors comment on the relative significance of these uncertainties to the nonideal solution effects calculated here?

- How often in the simulation are the activity coefficients recalculated, i.e. as concentrations change in the aqueous phase? Does it happen at every timestep?

- Since much depends in this study on the model chosen for activity coefficients, it would have been nice to see a sensitivity study of other popular approaches that could have been used instead. This is done somewhat in Table 3 and page, last paragraph. Given the sharp criticism presented, the discussion of Mao et al (2013) in this paragraph needs more elaboration (how do you know the implementation was 'incorrect'?), perhaps in the SI.

---

## Author Comment (AC1) · 25 Mar 2020

**"Treatment of non-ideality in the multiphase model SPACCIM-Part2: Impacts on the multiphase chemical processing in deliquesced aerosol particles"** *by* **Ahmad J. Rusumdar et al.**

**Anonymous Referee #1**

We thank reviewer 1 for the constructive comments to improve the manuscript. The comments of the reviewer are addressed point by point in the sections below. The answers to the reviewer comments are marked as blue text. Furthermore, all changes made in the manuscript are marked in the version with tracked changes. In order to provide a more conceive manuscript and for the sake of clarity, the subsection on the $C_3$ chemistry has been moved to the supplement.

The study presented in this paper aims to quantify the activity coefficients of species dissolved in deliquescent atmospheric particles. The approach is based on the modeling of two continental scenarios (urban, remote) simulating the chemical evolution in various phases (gas - particle - cloud droplets). The study examines the activity coefficients of inorganic (HOx, SOx, H2O2, metals) and organic (C1-C3) species. The effect of the non-ideal behavior of the species on the dynamic evolution of the system is evaluated by comparing the results obtained when the ideal behavior in the aqueous phase is imposed or not. For the scenarios tested, the budget of key species (S(VI), H2O2, organic acids ...) is substantially modified when the non-ideal behavior of the solution is considered. The results presented are relevant to this journal. The paper is well organized but the writing could likely be more concise.

I have a few comments (see below) that the authors might consider for submitting a revised article to ACP.

1. The description of the two scenarios considered in this study is inadequate. The authors refer to an article by Ervens et al. (2003) (page 7, line 24), but I did not find more detailed information in the article cited. Additional information is necessary to characterize the simulations carried out with the box model (initial conditions, emissions, speciation of organic species, deposition, NOx regime, NH3 level, etc.) and to weigh the simulated behavior of the species (e.g. the accumulation of S(VI) presented in fig.3, page 16 and reaching 90 µg/m3). The details of the simulation conditions could for example be added to the additional material.

Answer to the reviewer comment:
We thank the reviewer for this comment. Unfortunately, the link provided in Ervens et al. (2003) is not up-to-date anymore. Therefore, we have added a link to the CAPRAM webpage, in addition to Ervens et al. (2003). There, initial gas phase conditions, emissions and deposition are provided.

2. The paper examines the effect of the non-ideality of the solution on the condensed phase reactivity but does not discuss its effect on the gas/particle partitioning, as suggested for example page 12, line 4. No result of the non-ideal behavior is presented on the simulated aerosol mass concentrations, their compositions (e.g. inorganic vs organic), their liquid water content… . Adding a section presenting the effects on the overall aerosol properties (mass concentration, composition, water content in particular) would be useful to evaluate the sensitivity of this non-ideal behavior in models dealing with aerosols formation.

Answer to the reviewer comment:

The applied model calculates both the phase partitioning and the occurring multiphase chemistry simultaneously. The latter can substantially affect the budgets of the compounds. Hence, the predicted concentrations in either phase are a result of both the phase partitioning and the multiphase chemistry. Therefore, it is difficult here to discuss the effects of the non-ideality on the gas/particle partitioning and the applied multiphase chemical reactions individually. Due to the focus of the present study on the occurring multiphase chemistry we have not focused on the phase partitioning topic alone and have only cited literature (Cappa et al. (2008)) addressing this issue in more detail.

According to the reviewer comment, we have added a section to the SI presenting additional model results, e.g. total and organic aerosol mass for different model runs/scenarios.

3. The iron activity decreases notably when the non-ideal behavior of the species is considered. The authors show that this decrease in activity is one of the main causes explaining the simulated effects. However, the authors state that the model ignores the "middle range interaction" for Fe and Mg ions, leading to uncertainties about the activity coefficients calculated for these species. This limitation is reminded very regularly in the paper, including in the conclusion of the article (e.g. line 15, page 34). Although clearly mentioned, this uncertainty considerably weakens the conclusions of this study. I think that a test would be necessary to examine the sensitivity of the simulations to uncertainties about the activity of iron. An educated guess could perhaps be made to estimate the effect of the middle range interactions, e.g. using another TMI as a proxy.

Answer to the reviewer comment:

The authors fully agree with the reviewer that the missing middle range interaction parameters for Fe and Mg ions might weaken the conclusions of this study. The authors already thought about this issue quite a bit during the analysis of the modelled data. An educated guess to estimate those middle range interaction parameters was already discussed. For this reason, we have already compared available middle range interaction parameters of different metals. This comparison showed that also similar metals, for example $Mg^{2+}$ and $Ca^{2+}$ (same charge and main group in the periodic table), can be characterized by rather different middle range interaction parameters. Moreover, the use of the $Cu^{2+}$ interactions parameters for Fe and Mn ions can be also difficult as the chemistry of copper is known to be different from those of Fe and Mn. Nevertheless, according to the reviewer comment, we have included a new section in the SI comparing the base model runs with runs where the parameters for $Cu^{2+}$ have been applied for $Fe^{2+}$ and $Mn^{2+}$. A link to the sensitivity studies in the supplement is now included into the revised manuscript (see Sect. 3.4).

4. Minor comments: The term "flux" is used systematically by the authors to define the rate of processes. I think that using "flux" is inappropriate here and should be replaced by "rate" everywhere in the text and in figures 5, 8, 10 and 12.

Answer to the reviewer comment:
The term "flux(es)" has been replaced by "rate(s)" in the revised manuscript.

5. The writing of the chemical formula are often difficult to read for organic molecules. For example in Table 4, OH2CHCH2OH would be more easy to read as CH(OH)2CH2OH to represent hydrated glycolaldehyde. Similarly, C2H4COOH2 would be more easy to read as C(O)OHCH2CH2C(O)OH to represent succinic acid, etc . . . Furthermore, the writing is sometime misleading, like in figure 8 for OHCCHCHCHO (which I guess is for butenedial CHOCH=CHCHO) or, in figure 10, C(OH)2O2COOH which I cannot identify (a typo?) but

understand to be glyoxylic acid from the text. I suggest to use only "conventional" formula in the text, figures and tables.

Answer to the reviewer comment:
The writing of the chemical formulas has been revised.

Page 2, line 20: the word "including" in the context of the sentence is unclear. Change to "leading to"?
Answer to the reviewer comment:
"including" was replaced by "leading to".

Page3, line 16: the paper use both ALWC and ALW. Duplication of 2 similar acronyms could be avoided (ALWC is used only 2 times in paper).
Answer to the reviewer comment:
The ALWC is not used anymore in the revised manuscript.

Page 10, line 26-27: One of the 2 "finally" should be removed.
Answer to the reviewer comment:
One "finally" has been removed.

Page 11, line 22: pyruvic acid is not a simple carboxylic acid (i.e. without another functional group), as suggested in the context of the sentence.
Answer to the reviewer comment:
The sentence has been rephrased and reads now as follows:

"Only carboxylic acids with further substituents and a longer non-polar carbon chain (e.g. pyruvic and butyric acid) show higher activity coefficients, particularly with decreasing RH."

Page 12, line 18-20. The sentence seems useless in the context of the paragraph and could be removed.
Answer to the reviewer comment:
The sentence has been removed.

Page 13, line 8. I don't understand the meaning of the word "obstruct" in the context of the sentence.
Answer to the reviewer comment:
The sentence has been rephrased and reads now as follows.

"Mainly, microphysical parameters such as the ALW content pattern pH and ionic strength."

Page 14, line 5-6: the sentence is obvious in the context of the paragraph and could be removed.
Answer to the reviewer comment:
The sentence has been removed.

Page 15, line 14-16: The sentence is obvious in the context of the paragraph and could be removed.
Answer to the reviewer comment:
The sentence has been removed.

Page 19, line 1: the second processing in "processing of Fe(II) processing aqueous" could be removed.
Answer to the reviewer comment:
The second processing as been removed. The sentence reads now as follows:

"To summarise, the model simulations implicate that the multiphase processing of Fe(II) in aqueous particles is significantly affected by the treatment of non-ideality and the effect depends strongly on the ALW conditions."

Page 19, line 17-31: The reading of this paragraph is tedious. A figure similar to 5, 8, 10 or 12 could be useful.

Answer to the reviewer comment:

For the sake of clarity, we have not included such a Fig. here, but according to the reviewer comment, we have added a similar Fig. for $H_2O_2$ in the SI (see Fig. S6).

Page 21, line 7: units are missing for OH concentration. Number of significant figures should be consistent for the 2 numbers.

Answer to the reviewer comment:

The unit has been added and the numbers were unified.

Page 24, line 25: I don't understand "formation-substituted" in the sentence.

Answer to the reviewer comment:

The sentence has been rephrased as follows:

"…in-cloud oxidations of $C_2$ carbonyl compounds such as glycolaldehyde and glyoxal lead to the formation of substituted organic acids (e.g. glyoxylic and glycolic acid),…"

Page 26, line 10-32: The reading of this paragraph is tedious. A figure similar to 5, 8, 10 or 12 could be useful.

For the sake of clarity, we have not included such a Fig. here, but according to the reviewer comment, we have added similar Fig.s for glycolic and glyoxalic acid in the SI (see Fig. S7-8).

Page 31, line 8: Typo in "sSect."

Answer to the reviewer comment:

The typo was corrected.

Page 31, line 14: it is stated that keto malonic acid is the final C3 oxidation product. Figure 12 (page 32) show however a loss process which is not discussed in the text (and not "explained" from the legend in the figure).

Answer to the reviewer comment:

The comment is difficult to understand. Keto malonic acid is the $C_3$ oxidation product of this reaction chain, subsequent products only include 2 carbon atoms. According to the reviewer comment, the word has been removed from the sentence.

Page 32, figure 12: Only the first 3 processes (dark blue, light blue and pink) can be discerned in the figure. The other processes could be merged in either the "other source" or "other sinks" boxes.

Answer to the reviewer comment:

The Figure has been revised.

**References:**

Ali, H. M., Iedema, M., Yu, X. Y., and Cowin, J. P.: Ionic strength dependence of the oxidation of $SO_2$ by $H_2O_2$ in sodium chloride particles, Atmos. Environ., 89, 731-738, https://doi.org/10.1016/j.atmosenv.2014.02.045, 2014.

Cappa, C. D., Lovejoy, E. R., and Ravishankara, A. R.: Evidence for liquid-like and nonideal behavior of a mixture of organic aerosol components, P. Natl. Acad. Sci. USA, 105, 18687-18691, https://doi.org/10.1073/pnas.0802144105, 2008.

Cheng, Y., Zheng, G., Wei, C., Mu, Q., Zheng, B., Wang, Z., Gao, M., Zhang, Q., He, K., Carmichael, G., Poschl, U., and Su, H.: Reactive nitrogen chemistry in aerosol water as a source of sulfate during haze events in China, Sci. Adv., 2, e1601530, https://doi.org/10.1126/sciadv.1601530, 2016.

Deguillaume, L., Leriche, M., Desboeufs, K., Mailhot, G., George, C., and Chaumerliac, N.: Transition metals in atmospheric liquid phases: Sources, reactivity, and sensitive parameters, Chem. Rev., 105, 3388-3431, 2005.

Herrmann, H., Schaefer, T., Tilgner, A., Styler, S. A., Weller, C., Teich, M., and Otto, T.: Tropospheric aqueous-phase chemistry: Kinetics, mechanisms, and its coupling to a changing gas phase, 115, 4259-4334, https://doi.org/doi:10.1021/cr500447k, 2015.

Lagrange, J., Pallares, C., Wenger, G., and Lagrange, P.: Electrolyte effects on aqueous atmospheric oxidation of sulphur dioxide by hydrogen peroxide, Atmos. Environ., 27, 129-137, https://doi.org/https://doi.org/10.1016/0960-1686(93)90342-V, 1993.

Lagrange, J., Pallares, C., and Lagrange, P.: Electrolyte effects on aqueous atmospheric oxidation of sulphur dioxide by ozone, J. Geophys. Res.-Atmos., 99, 14595-14600, https://doi.org/10.1029/94JD00573, 1994.

Maaß, F., Elias, H., and Wannowius, K. J.: Kinetics of the oxidation of hydrogen sulfite by hydrogen peroxide in aqueous solution: ionic strength effects and temperature dependence, Atmos. Environ., 33, 4413-4419, https://doi.org/https://doi.org/10.1016/S1352-2310(99)00212-5, 1999.

Mao, J., Fan, S., Jacob, D. J., and Travis, K. R.: Radical loss in the atmosphere from Cu-Fe redox coupling in aerosols, 13, 509-519, https://doi.org/10.5194/acp-13-509-2013, 2013.

Martin, L. R., and Hill, M. W.: The effect of ionic strength on the manganese catalyzed oxidation of sulfur(IV), Atmos. Environ., 21, 2267-2270, https://doi.org/10.1016/0004-6981(87)90361-1, 1987.

Millero, F. J., and Woosley, R.: The Hydrolysis of Al(III) in NaCl solutions-A Model for Fe(III), Environ. Sci. Technol., 43, 1818-1823, https://doi.org/10.1021/es802504u, 2009.

Weller, C., Hoffmann, D., Schaefer, T., and Herrmann, H.: Temperature and ionic strength dependence of $NO_3$-radical reactions with substituted phenols in aqueous solution, 224, 1261-1287, https://doi.org/doi:10.1524/zpch.2010.6151, 2010.

---

## Author Comment (AC2) · 25 Mar 2020

**"Treatment of non-ideality in the multiphase model SPACCIM-Part2: Impacts on the multiphase chemical processing in deliquesced aerosol particles"** *by* **Ahmad J. Rusumdar et al.**

**Anonymous Referee #2**

The authors present a sensitivity study of the impact of considering non-ideal solution effects on predictions of aqueous aerosol chemistry using the SPACCIM-SpactMod model. The study is well-conceived, and the results highlight the potential significance of non-ideal solution effects in multiphase atmospheric chemistry, something that is generally ignored in atmospheric models due in part to computational expense but also because of a lack of data. I think the manuscript will be suitable for publication in ACP after a few clarifications.

The authors thank reviewer 2 for the positive and thoughtful comments to improve the manuscript. The comments of the reviewer are carefully addressed point by point in the section below. The answers to the reviewer comments are marked as blue text. Moreover, all changes made in the manuscript are marked in the version with tracked changes.

- The aqueous chemical kinetic data in CAPRAM, which the multiphase chemical kinetic simulations in this study were based on, for the most part was not originally measured/reported as a function of activity but rather of concentration. The measurements that form the basis of that data were also mostly performed under much more dilute and closer to ideal conditions than the aqueous aerosol conditions considered here. How accurate is it to use those rate laws but plug in the activities calculated here - which are in many cases very different from the activities where the rates were originally measured both because of the high concentrations of atmospheric aerosols but also because of the non-ideality? Is it even possible to evaluate this?

Answer to the reviewer comment:
The reviewer is right that most of the kinetic reaction parameters are determined in the laboratory under much more dilute and closer to ideal conditions than aqueous aerosol conditions. Thus, effects of important parameters such as the ionic strength are not investigated by such studies and are therefore not considered in the current version of CAPRAM. However, the ionic strength is a main parameter in the calculation of the activity coefficients and consequently also in the calculation of the reactive fluxes.

Ionic strength effects are believed to be an important parameter of particle chemical reactions such as S(IV) oxidation (Martin and Hill, 1987; Lagrange et al., 1993; Lagrange et al., 1994; Maaß et al., 1999; Ali et al., 2014; Cheng et al., 2016), although experimental data at the extremely high ionic strengths typical of atmospheric aerosols are limited. Studies available studying the ionic strength effects (see e.g. Weller et al. (2010) and Herrmann et al. (2015) and references therein), however, often only quantify the overall effect of both ionic strength and the activity coefficients. To provide ionic-strength dependent reaction rate constants, the activity coefficients of the educts need to be considered to finally derive the single effect of the ionic strength. Such combined laboratory studies and model calculations would be a way to come up with datasets that can overcome the current model implementation limitation.

The current limitation in the applied mechanism is now briefly addressed in the revised manuscript in Sect.2.2. and reads as follows:

"Overall, it is worth to be noted that most of the kinetic reaction parameters considered in CAPRAM are determined in the laboratory under dilute and closer to ideal conditions rather

than concentrated aqueous aerosol conditions. Thus, effects of important concentrated solution parameters such as ionic strength have not been investigated by such studies and are therefore not considered in the current version of CAPRAM. Once more ionic-strength dependent reaction rate constants become available, they have to be considered in future mechanisms."

- I agree with the other referee's comment that the uncertainty in the MR parameterizations for Fe and Mn are a significant weakness, especially given the likely importance of those two ions in multiphase atmospheric chemistry. Could the authors estimate those parameters, perhaps from available lab data, to at least constrain the effect?

Answer to the reviewer comment:
The authors fully agree with the reviewer that the missing middle range interaction parameters for Fe and Mg ions might weaken the conclusions of this study. The authors already thought about this issue quite a bit during the analysis of the modelled data. An educated guess to estimate those middle range interaction parameters was already discussed. For this reason, we have already compared available middle range interaction parameters of different metals. This comparison showed that also similar metals, for example $Mg^{2+}$ and $Ca^{2+}$ (same charge and main group in the periodic table), can be characterized by rather different middle range interaction parameters. Moreover, the use of the $Cu^{2+}$ interactions parameters for Fe and Mn ions can be also difficult as the chemistry of copper is known to be different from those of Fe and Mn. Nevertheless, according to the reviewer comment, we have included a new section in the SI comparing the base model runs with runs where the parameters for $Cu^{2+}$ have been applied for $Fe^{2+}$ and $Mn^{2+}$. A link to the sensitivity studies in the supplement is now included into the revised manuscript (see Sect. 3.4).

- Considerable uncertainty exists in the measured and reported rate data for much TMI chemistry, especially when it comes to ionic strength and pH dependence. Can the authors comment on the relative significance of these uncertainties to the non-ideal solution effects calculated here?

Answer to the reviewer comment:
The reviewer is certainly right that there are considerable uncertainties in available reaction rate data of the TMI chemistry and potential gaps exists in the scientific knowledge about their speciation, reactivity and complex interactions. The uncertainties and knowledge gaps are surely related to the complex dependencies of the TMI chemistry on the acidity and ionic strength (see Deguillaume et al. (2005) and references therein) and are able to affect different key chemical subsystems such as the HOx, organic and sulfur chemistry. CAPRAM contains already a rather complex TMI chemistry implementation which is far from being complete due to many gaps in the scientific knowledge. When better measured chemical rate constants of TMI reactions become available they definitely need to be integrated into upcoming CAPRAM mechanism versions.

Regarding the relative significance of the uncertainties compared to the non-ideal solution effects, the authors think that a more advanced TMI chemistry knowledge would be surely needed but effects of non-ideality will be definitely also play a key role due to the potential impact of activity coefficients on the reaction rates and speciation constants. Thus, laboratory future studies need to investigate important parameters such as the "middle range interaction" parameters of different TMI compounds as they can impact the overall rates by about 2 orders of magnitude depending on the non-ideality of the solution.

- How often in the simulation are the activity coefficients recalculated, i.e. as concentrations change in the aqueous phase? Does it happen at every timestep?

Answer to the reviewer comment:
In simulation, activity coefficients are calculated at each timestep. Further details about the implemented procedure is given in Rusumdar et al. (2016). According to the reviewer comment, we have slightly updated SPACCIM-SpactMod description in Sect. 2.1 and address this issue now in the revised manuscript.

- Since much depends in this study on the model chosen for activity coefficients, it would have been nice to see a sensitivity study of other popular approaches that could have been used instead. This is done somewhat in Table 3 and page, last paragraph. Given the sharp criticism presented, the discussion of Mao et al (2013) in this paragraph needs more elaboration (how do you know the implementation was 'incorrect'?), perhaps in the SI.

Answer to the reviewer comment:

1. A sensitivity study using other popular activity coefficient approaches (EAIM, …) that could have been applied instead of AIOMFAC is an excellent idea but beyond the scope of the present study. A comprehensive sensitivity study using other popular activity coefficient approaches will be definitely a gainful task for the future.

2. Unfortunately, most multiphase chemistry studies do not apply activity coefficients in their models or, if activity coefficients are considered, studies do often not provide the calculated activity coefficients in the publications. We did again a literature survey and have extended Table 3 with available data found and updated the discussion in Sect. 3.1.1. Nevertheless, presently no comprehensive comparison is possible.

3. In the study of Mao et al. (2013), an activity coefficient $\gamma$ of 0.01 was applied for $Fe^{3+}$ based on the lowest estimate from Millero and Woosley (2009). However, the lowest estimate in Millero and Woosley (2009) is $\ln(\gamma)=-2$, i.e. $\gamma$ should be 0.135. Thus, the decadic logarithm was used by Mao et al. (2013) instead of the natural logarithm. This issue is now outlined in the remarks of Table 3.

**References:**

Ali, H. M., Iedema, M., Yu, X. Y., and Cowin, J. P.: Ionic strength dependence of the oxidation of SO2 by H2O2 in sodium chloride particles, Atmos. Environ., 89, 731-738, https://doi.org/10.1016/j.atmosenv.2014.02.045, 2014.

Cappa, C. D., Lovejoy, E. R., and Ravishankara, A. R.: Evidence for liquid-like and nonideal behavior of a mixture of organic aerosol components, P. Natl. Acad. Sci. USA, 105, 18687-18691, https://doi.org/10.1073/pnas.0802144105, 2008.

Cheng, Y., Zheng, G., Wei, C., Mu, Q., Zheng, B., Wang, Z., Gao, M., Zhang, Q., He, K., Carmichael, G., Poschl, U., and Su, H.: Reactive nitrogen chemistry in aerosol water as a source of sulfate during haze events in China, Sci. Adv., 2, e1601530, https://doi.org/10.1126/sciadv.1601530, 2016.

Deguillaume, L., Leriche, M., Desboeufs, K., Mailhot, G., George, C., and Chaumerliac, N.: Transition metals in atmospheric liquid phases: Sources, reactivity, and sensitive parameters, Chem. Rev., 105, 3388-3431, 2005.

Herrmann, H., Schaefer, T., Tilgner, A., Styler, S. A., Weller, C., Teich, M., and Otto, T.: Tropospheric aqueous-phase chemistry: Kinetics, mechanisms, and its coupling to a changing gas phase, 115, 4259-4334, https://doi.org/doi:10.1021/cr500447k, 2015.

Lagrange, J., Pallares, C., Wenger, G., and Lagrange, P.: Electrolyte effects on aqueous atmospheric oxidation of sulphur dioxide by hydrogen peroxide, Atmos. Environ., 27, 129-137, https://doi.org/https://doi.org/10.1016/0960-1686(93)90342-V, 1993.

Lagrange, J., Pallares, C., and Lagrange, P.: Electrolyte effects on aqueous atmospheric oxidation of sulphur dioxide by ozone, J. Geophys. Res.-Atmos., 99, 14595-14600, https://doi.org/10.1029/94JD00573, 1994.

Maaß, F., Elias, H., and Wannowius, K. J.: Kinetics of the oxidation of hydrogen sulfite by hydrogen peroxide in aqueous solution: ionic strength effects and temperature dependence, Atmos. Environ., 33, 4413-4419, https://doi.org/https://doi.org/10.1016/S1352-2310(99)00212-5, 1999.

Mao, J., Fan, S., Jacob, D. J., and Travis, K. R.: Radical loss in the atmosphere from Cu-Fe redox coupling in aerosols, 13, 509-519, https://doi.org/10.5194/acp-13-509-2013, 2013.

Martin, L. R., and Hill, M. W.: The effect of ionic strength on the manganese catalyzed oxidation of sulfur(IV), Atmos. Environ., 21, 2267-2270, https://doi.org/10.1016/0004-6981(87)90361-1, 1987.

Millero, F. J., and Woosley, R.: The Hydrolysis of Al(III) in NaCl solutions-A Model for Fe(III), Environ. Sci. Technol., 43, 1818-1823, https://doi.org/10.1021/es802504u, 2009.

Weller, C., Hoffmann, D., Schaefer, T., and Herrmann, H.: Temperature and ionic strength dependence of NO3-radical reactions with substituted phenols in aqueous solution, 224, 1261-1287, https://doi.org/doi:10.1524/zpch.2010.6151, 2010.

---

## Author Comment (AC3) · 25 Mar 2020

The comment was uploaded in the form of a supplement:
https://www.atmos-chem-phys-discuss.net/acp-2019-819/acp-2019-819-AC3-supplement.pdf

---

## Author Comment (AC4) · 25 Mar 2020

[revised manuscript text omitted]

**Appendix B: Figures**

[Figure]

**Figure S3.** Schematic of multiphase mechanism employed in this study, including the number of processes, reactions, and phase transfer processes (modified from Deguillaume et al. (2009); Tilgner and Herrmann (2010)).

[Figure]

**Figure S4.** Modelled Liquid Water Content in $l_{(water)}$ m$^{-3}_{(air)}$ throughout the simulation time for the different urban (left) and remote (right) simulation cases (90 %-IDR/90 %-NIDR and 70 %-IDR/70 %-NIDR).

[Figure]

**Figure S5.** Modelled concentrations in µg m$^{-3}_{(air)}$ of the total (top), and organic (bottom) aerosol mass throughout the simulation time for the different remote simulation cases (90 %-IDR/90 %-NIDR and 70 %-IDR/70 %-NIDR).

**Additional reaction rate analyses plots**

[Figure]

**Figure S6** Modelled chemical sink and source mass rates of $H_2O_2$ in the aqueous phase in mol m$^{-3}$ s$^{-1}$ for the second day of modelling time for the urban scenario for the simulations 90 %-IDU vs. 90 %-NIDU. a) Ideal solutions (90 %-IDU), b) non-ideal solutions (90 %-NIDU), c) corresponding total rates.

[Figure]

**Figure S7** Modelled chemical sink and source mass rates of glycolic acid in the aqueous phase in mol m⁻³ s⁻¹ for the second day of modelling time for the urban scenario for the simulations 90 %-IDU vs. 90 %-NIDU. a) Ideal solutions (90 %-IDU), b) non-ideal solutions (90 %-NIDU), c) corresponding total rates.

[Figure]

**Figure S8**    Modelled chemical sink and source mass rates of glyoxylic acid in the aqueous phase in mol m$^{-3}$ s$^{-1}$ for the second day of modelling time for the urban scenario for the simulations 90 %-IDU vs. 90 %-NIDU. a) Ideal solutions (90 %-IDU), b) non-ideal solutions (90 %-NIDU), c) corresponding total rates.

**Model results of the remote simulation cases**

[Figure]

**Figure S9.** Modelled pH value (left) and ionic strength (I, right) as a function of simulation time for the different remote simulation cases (90 %-IDR/90 %-NIDR and 70 %-IDR/70 %-NIDR).

[Figure]

**Figure S10.** Modelled Fe(II) aqueous-phase concentration in ng m$^{-3}$ throughout the modelling time (left) and corresponding time evolution of activity coefficients (right) for the different remote simulation cases (90 %-IDR/90 %-NIDR and 70 %-IDR/70 %-NIDR).

[Figure]

**Figure S11.** Modelled gas- and aqueous-phase concentration of $H_2O_2$ throughout the simulation time for the different remote simulation cases (90 %-IDR/90 %-NIDR and 70 %-IDR/70 %-NIDR).

[Figure]

**Figure S12.** Modelled aqueous-phase OH concentration in mol $l^{-1}$ throughout the simulation time for the different remote simulation cases (90 %-IDR/90 %-NIDR and 70 %-IDR/70 %-NIDR).

[Figure]

**Figure S13**. Modelled aqueous-phase concentrations in ng m$^{-3}$$_{(air)}$ and corresponding activity coefficients for important C$_2$ oxidation products, (i) glycolic acid (top), (ii) glyoxylic acid (centre), and (iii) oxalic acid (down) throughout the simulation time for the different remote simulation cases (90 %-IDR/90 %-NIDR and 70 %-IDR/70 %-NIDR). The plotted concentrations represent the sum of dissociated and undissociated forms of the carboxylic acids.

[Figure]

**Figure S14.** Modelled aqueous-phase concentrations in ng m$^{-3}$(air) and corresponding activity coefficients for important C$_3$ oxidation products, (i) pyruvic acid (top), (ii) 3-oxo-pyruvic acid (centre), and (iii) keto malonic acid (down) throughout the simulation time for the different remote simulation cases (90 %-IDR/90 %-NIDR and 70 %-IDR/70 %-NIDR). The plotted concentrations represent the sum of dissociated and undissociated forms of the carboxylic acids.

**Appendix C: Additional sensitivity runs**

In order to study, the sensitivity of the model with regards to missing middle-range (MR) interaction parameters, additional sensitivity simulations have been performed (acronyms: 90/70% NIDU/NIDR (MR)). For the sensitivity simulations, the MR interaction parameters of $Cu^{2+}$ has been applied for $Fe^{2+}/Mn^{2+}$. As there are no MR interaction parameters for triply charged transition metal ions available, no MR values are applied for those transition metal ions. However, since particularly $Fe^{2+}$ is the key transition metal ion for the overall TMI processing, the performed sensitivity studies reveal the potential effects of an improved description of the MR interaction parameters.

In Fig. S13 and S14, simulation results of non-ideal runs with and without considered MR interaction parameters are shown for key chemical substances that are sensitive to a non-ideal treatment under remote and urban conditions. The consideration of MR interaction parameters for $Fe^{2+}/Mn^{2+}$ leads to similar activity coefficient pattern for $Cu^{2+}$ and $Fe^{2+}/Mn^{2+}$ (cp. Fig. S15). In comparison to base run without the MR parameters, the activity coefficient values of $Fe^{2+}/Mn^{2+}$ significantly higher and even above 1 under 70% RH conditions as observed for $Cu^{2+}$. Particularly, the higher activity coefficient values of $Fe^{2+}$ in the MR-cases significantly affects key chemical subsystems (cp. Fig. S16 and S17). For $H_2O_2$, the model results show that the consideration of MR parameters leads to lower concentrations levels due to more effective loss processes such as the Fenton chemistry. As a consequence, the aqueous OH concentrations are significantly raised and slightly higher oxalic acid concentrations are modelled indicating a more effective organic acid formation as a result of the higher OH levels. On the other hand, the effects on the S(VI) formations are relatively small and only present in the urban simulations.

Because of the higher activity coefficient under consideration of MR parameters, the model results shift at least to a certain part towards the simulations without non-ideal chemistry treatment with a more active TMI-HOx cycling under deliquesced aerosol conditions. Overall, the sensitivity studies clearly show that middle-range (MR) interaction parameters of key TMIs represents crucial parameter that need to be determined in future laboratory experiments to improve current model implementations.

[Figure]

**Figure S15.** Modelled activity coefficients of $Fe^{2+}$ (left) and $Cu^{2+}$ (right) with and without considered MR parameters for the different non-ideal urban (top) and remote (bottom) model cases.

[Figure]

**Figure S16.** Modelled aqueous-phase concentrations of aqueous OH radical (top left), H$_2$O$_2$ (top right), sulfur(VI) (bottom left) and oxalic acid (bottom right) with and without considered MR parameters for the different non-ideal urban model cases.

[Figure]

**Figure S17.** Modelled aqueous-phase concentrations of aqueous OH radical (top left), H₂O₂ (top right), sulfur(VI) (bottom left) and oxalic acid (bottom right) with and without considered MR parameters for the different non-ideal remote model cases.

**Appendix D: Tables**

**Table S1.** Initial chemical aerosol particle composition (relative contributions to the total particulate non-water mass) for the urban and remote environmental model scenarios.

| Compound | Urban | Remote |
|---|---|---|
| $NH_4^+$ | 1.00E-01 | 9.80E-02 |
| $NO_3^-$ | 2.49E-02 | 9.27E-02 |
| $SO_4^{2-}$ | 2.61E-01 | 1.91E-01 |
| $Cl^-$ | 1.92E-02 | 9.70E-05 |
| $Br^-$ | 6.53E-04 | 1.77E-04 |
| $I^-$ | 1.23E-04 | 1.52E-07 |
| $Mn^{3+}$ | 1.54E-04 | 1.15E-04 |
| $Fe^{3+}$ | 1.46E-03 | 1.42E-03 |
| $Cu^{2+}$ | 1.53E-04 | 1.08E-04 |
| *WSOM* | 1.51E-01 | 5.57E-02 |
| $HC_2O_4^-$ | 4.38E-03 | 1.62E-03 |
| $HOOCCH_2COO^-$ | 2.79E-03 | 1.03E-03 |
| $HOOCC_2H_4COO^-$ | 1.59E-03 | 5.89E-04 |
| *WISOM* | 2.39E-01 | 8.83E-02 |
| *EC* | 1.56E-02 | 1.67E-01 |
| *Other anions* | 7.96E-05 | 1.89E-04 |
| *Cations (+)* | 3.26E-02 | 1.53E-02 |
| *Cations (2+)* | 1.49E-02 | 1.85E-02 |
| *Other metals* | 3.25E-02 | 4.05E-02 |
| *SiO2* | 9.53E-02 | 2.27E-01 |
| *P* | 2.87E-03 | 2.73E-04 |

**Remarks:** Single species and compound groups marked in italics are treated in SPACCIM as non-reactive species. The respective ions are just considered for the charge balance.

**Table S2.** Parameters (N: Number, ρ: Density, r: Radius) of the mono-disperse aerosol particle initialisation for the urban and remote environmental model scenarios.

| Parameter | Urban | Remote |
|---|---|---|
| N (#/cm$^3$) | 7.0E+08 | 1.0E+08 |
| ρ (kg/m$^3$) | 1770 | 1770 |
| r (m) | 2.0E+07 | 2.0E+07 |